# Loss of β-cell identity and diabetic phenotype in mice caused by disruption of CNOT3-dependent mRNA deadenylation

Dina Mostafa [1,2], Akiko Yanagiya [1], Eleni Georgiadou [3], Yibo Wu[4], Theodoros Stylianides[5], Guy A. Rutter [3], Toru Suzuki [6✉] & Tadashi Yamamoto [1✉]

Pancreatic β-cells are responsible for production and secretion of insulin in response to increasing blood glucose levels. Defects in β-cell function lead to hyperglycemia and diabetes mellitus. Here, we show that CNOT3, a CCR4–NOT deadenylase complex subunit, is dysregulated in islets in diabetic *db/db* mice, and that it is essential for murine β cell maturation and identity. Mice with β cell-specific *Cnot3* deletion (*Cnot3*βKO) exhibit impaired glucose tolerance, decreased β cell mass, and they gradually develop diabetes. *Cnot3*βKO islets display decreased expression of key regulators of β cell maturation and function. Moreover, they show an increase of progenitor cell markers, β cell-disallowed genes, and genes relevant to altered β cell function. *Cnot3*βKO islets exhibit altered deadenylation and increased mRNA stability, partly accounting for the increased expression of those genes. Together, these data reveal that CNOT3-mediated mRNA deadenylation and decay constitute previously unsuspected post-transcriptional mechanisms essential for β cell identity.

[1] Cell Signal Unit, Okinawa Institute of Science and Technology Graduate University, Okinawa, Japan. [2] Department of Biochemistry, Faculty of Pharmacy, Ain Shams University, Cairo, Egypt. [3] Section of Cell Biology and Functional Genomics, Department of Metabolism, Digestion and Reproduction, Imperial College London, Hammersmith Hospital, London, UK. [4] Laboratory for Next-Generation Proteomics, Riken Center of Integrative Medical Sciences, Yokohama, Japan. [5] Centre of Innovative and Collaborative Construction Engineering, Loughborough University, Leicestershire, UK. [6] Laboratory for Immunogenetics, Riken Center of Integrative Medical Sciences, Yokohama, Japan. ✉email: toru.suzuki.ff@riken.jp; tadashi.yamamoto@oist.jp

Diabetes mellitus is a metabolic disorder that is ultimately caused by the loss of functional pancreatic β cells[1]. Type 1 diabetes (T1D) involves immune-mediated β-cell destruction[1], but dedifferentiation and dysfunction, rather than β-cell loss per se, are implicated in the pathogenesis of type 2 diabetes (T2D)[2]. A fuller understanding of molecular mechanisms that govern β-cell differentiation, maturation, and dedifferentiation in T2D will likely prove crucial for developing novel prevention and treatment strategies.

Pancreatic β cells are functionally defined by their capacity for insulin secretion, stimulated by glucose and other nutrients[3]. Underlying this unique function is a molecular identity defined by expression of genes that drive β-cell maturation, insulin gene expression, insulin granule formation, and exocytosis. Although extensive research has focused on transcriptional and epigenetic regulation of β-cell maturation and function, whether and how post-transcriptional mechanisms influence β-cell gene expression program have been little explored. Nonetheless, from pre-existing mRNAs, glucose stimulates biosynthesis of insulin and other β-cell proteins, particularly those involved in insulin granule biogenesis[4]. Furthermore, long non-coding RNAs (lncRNAs) and micro-RNAs (miRNAs) act as important modulators of pancreatic β-cell gene expression and function[5–7].

Most eukaryotic mRNAs are polyadenylated at the 3′ end, and poly(A) tail lengths are important for post-transcriptional gene regulation[8]. Importantly, mRNA deadenylation is the rate-limiting step in mRNA decay[9]. Deadenylation shortens poly(A) tail lengths, reducing the binding of cytoplasmic polyadenosine-binding protein 1. This decreases mRNA stability, slowing translation[8]. The CCR4–NOT complex constitutes the major deadenylase complex in eukaryotes[10,11]. CCR4–NOT is a multimeric complex containing two catalytic subunits with deadenylase activity (CNOT6 or CNOT6L, and CNOT7 or CNOT8) and six non-catalytic subunits (CNOT1, CNOT2, CNOT3, CNOT9, CNOT10, and CNOT11)[11]. Although the exact functions of the non-catalytic subunits remain elusive, they may regulate stability and activity of the catalytic subunits and may participate in recruitment of the CCR4–NOT complex to specific target mRNAs[12]. Two targeting mechanisms have been proposed to ensure specificity. First, sequence-specific RNA-binding proteins (RBPs) bring the CCR4–NOT complex to sequence elements in the 3′ untranslated region of the target mRNA, and second, the miRNA machinery recruits the CCR4–NOT complex to its target mRNA[13–15]. Earlier studies have revealed the involvement of the CCR4–NOT complex in development of metabolic disorders such as obesity, diabetes, and lipodystrophy;[16–18] however, its role in maintaining normal β-cell function has not been investigated.

In this study, we examined whether and how mRNA deadenylation regulates β-cell function by analyzing the physiological role of the CCR4–NOT complex in pancreatic β cells. We focused on the CNOT3 subunit and we present evidence that CNOT3-dependent deadenylation is pivotal for normal β-cell function. We show that β-cell-specific knockout (KO) of Cnot3 results in impaired glucose tolerance and ultimately the development of overt diabetes. This might be attributable to a lack of β-cell identity. Our data suggest that the observed β-cell dysfunction can be partly explained by a loss of CNOT3-dependent control of the decay of Aldob, Slc5a10, Wnt5b, and several other mRNAs that are normally suppressed in β cells[19-21]. Thus, we propose that CNOT3 is involved in degrading mRNAs from these genes to maintain normal β-cell function. Our findings show that the CCR4–NOT complex is deregulated in pancreatic islets in diabetes, thus suggesting that the CCR4–NOT complex serves as a therapeutic target to treat diabetes.

## Results

**CNOT3 decreases in diabetic and gluco/lipo-toxic conditions.** We first asked whether CCR4–NOT complex subunit expression is altered in the diabetic state. Accordingly, we isolated islets from *db/db* mice, which lack the leptin receptor and develop severe obesity associated with diabetes[22]. Immunoblot analysis revealed a decrease in CCR4–NOT complex subunits, CNOT1, CNOT2, and CNOT3 (Fig. 1a and Supplementary Fig. 1a, 2a) in diabetic islets. Among these subunits, CNOT3 consistently showed a marginally significant decrease in all samples examined (Supplementary Fig. 1a and 2a). Since CNOT3 is an important subunit of the CCR4–NOT complex[17], these data suggest impaired CCR4–NOT complex function in diabetic islets. To investigate whether CCR4–NOT is a possible early effector in the pathogenesis of diabetes, we examined CCR4–NOT complex subunit expression in the prediabetic state using 20-week-old mice fed a high-fat diet (HFD) for 3 months. We observed a significant increase of CNOT8 (Fig. 1b and Supplementary Fig. 1b, 2b). In order to determine whether these effects on CCR4–NOT complex subunits were the result of gluco/lipotoxicity, we analyzed CCR4–NOT subunit expression in MIN6 cells after chronic exposure (1 week) to high glucose (50 mM), with or without palmitic acid (500 μM). CNOT3 significantly decreased with high glucose and palmitic acid treatments (Fig. 1c, Supplementary Fig. 1c, 2c). CNOT8 increased with palmitic acid treatment in all samples examined, although the extent of the increase varied among samples (Fig. 1c, Supplementary Fig. 1c, 2c).

**Impaired insulin secretion in *Cnot3*βKO mice.** In light of the above findings, we next investigated roles of the CCR4–NOT complex in pancreatic β cells. We generated a mouse model lacking the *Cnot3* gene in β cells (*Cnot3*βKO mice). Successful suppression of CNOT3 in β cells was confirmed by immunoblot analysis and immunohistochemistry (Fig. 2a, b). Immunohistochemistry revealed CNOT3 suppression in centrally located β cells, while other islet cells located peripherally retained CNOT3 expression in *Cnot3*βKO islets. In mice, β cells make up the core of the islets[23], suggesting that CNOT3-expressing cells in *Cnot3*βKO islets are non-β cells. Indeed, cells that did not undergo Cre-mediated recombination, expressed glucagon (GLUC), somatostatin (SST), and pancreatic polypeptide (PPT), but not insulin, as shown in Fig. 3g. Consistent with CNOT3-suppression in other tissues[24–26], the abundance of other complex subunits, such as CNOT1 and CNOT2, decreased in *Cnot3*βKO islets (Supplementary Fig. 3a, b). Moreover, bulk poly(A) tail analysis showed that the population of mRNAs with poly(A) tail lengths longer than 50 nucleotides increased in *Cnot3*βKO islets, compared with controls (Fig. 2c, d), indicating impairment of deadenylase activity.

To determine whether CNOT3 depletion affects β-cell function, we conducted glucose tolerance tests (GTT) on control and *Cnot3*βKO mice. We used 4- and 8-week-old mice. Four-week-old *Cnot3*βKO mice showed glucose tolerance and fasting blood glucose levels comparable to those of control mice (Fig. 2e). While fasting blood glucose levels were still similar between 8-week-old controls and *Cnot3*βKO mice, *Cnot3*βKO mice showed impaired glucose tolerance (Fig. 2f). By the time *Cnot3*βKO mice reached 12 weeks of age, they had developed overt diabetes, as indicated by significant elevation of fasting blood glucose levels (Fig. 2g). With development of overt diabetes, 12-week-old *Cnot3*βKO mice displayed general deterioration of health and clinical signs of diabetes, including polyuria, polydipsia, and weight loss (Fig. 2h). These results suggest that metabolic impairment worsened with age, supporting progressive loss of

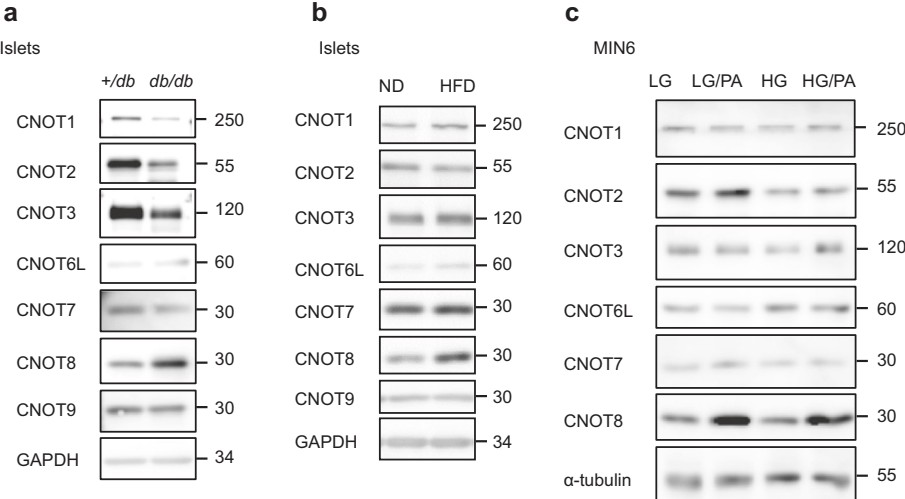

**Fig. 1 CCR4–NOT complex subunits are deregulated in mouse models of diabetes and obesity. a–c** Immunoblot analysis of CCR4–NOT complex subunits in: **a** islet lysates from 16-week-old +/db control and db/db mice. **b** islet lysates from 20-week-old mice fed a normal diet (ND) or a high-fat diet (HFD) for 12 weeks. **c** MIN6 cells under low/high-glucose conditions (LG/HG) with or without palmitic acid (PA) treatment. Each blot is a representative of three different blots.

β-cell function in Cnot3βKO mice. Indeed, blood glucose levels in 8-week-old Cnot3βKO mice were significantly elevated above those of control mice under normal feeding conditions (Fig. 2i). Furthermore, we measured fasting serum insulin levels and serum insulin levels after glucose administration in 8-week-old mice. There was no significant difference in fasting serum insulin levels between Cnot3βKO mice and controls. However, Cnot3βKO mice had significantly lower serum insulin levels 15 min after glucose administration, compared with controls (Fig. 2j). To evaluate β-cell function, we performed a glucose-stimulated insulin secretion (GSIS) assay on islets isolated from 8-week-old Cnot3βKO mice and controls. Cnot3βKO islets displayed normal basal insulin secretion at 3 mM glucose, but significantly decreased insulin secretion upon stimulation with high (17 mM) glucose, compared with control islets (Fig. 2k). This provides evidence for functional impairment of Cnot3βKO islets. Results of this ex vivo assay also agree with observed normal fasting serum insulin levels in Cnot3βKO mice and decreased serum insulin after in vivo glucose stimulation (Fig. 2j). Correspondingly, Cnot3βKO islets displayed defective glucose-stimulated cytosolic $Ca^{2+}$ dynamics, with a markedly more rapid return of cytosolic $Ca^{2+}$ to pre-stimulatory levels in Cnot3βKO than in control islets (Fig. 2l, m). In contrast, the fluorescent response of Cnot3βKO islets to depolarization with KCl was marginally augmented from that of control islets, consistent with the idea that observed β-cell dysfunction is largely due to defects in glucose metabolism (Fig. 2n). Intercellular β–β cell connectivity, adopting the Pearson correlation (R) analysis[27] was not altered by Cnot3 deletion, when perfused with either high glucose (17 mM) or KCl. Nevertheless, at low (3 mM) glucose, the number of possible β cell–β cell connections decreased in Cnot3βKO compared with control islets (Supplementary Fig. 4a, b). No differences in connectivity strength between groups were observed in response to high glucose or KCl (Supplementary Fig. 4c, d).

Evaluation of 8-week-old Cnot3βKO mouse islet morphology with H&E staining revealed pale nuclear staining with many islets showing abnormal shapes. At 12 weeks of age, almost all Cnot3βKO islets displayed abnormal shapes and neovascularization. At 20 weeks of age, Cnot3βKO islets had reduced numbers of cells in individual islets in a manner similar to that observed in diabetic Goto-Kakizaki rats[28] (Supplementary Fig. 4e).

**Reduced number of insulin-producing cells in Cnot3βKO mice.** β-cell-specific depletion of Cnot3 led to a significant reduction in expression of murine insulin gene isoforms (Ins1 and Ins2) (Fig. 3a). In order to trace Cnot3 depletion in β cells, we used Ins1-Cre mTmG reporter mice, in which successfully recombined cells show green fluorescence from expression of membrane-targeted EGFP (mG), whereas unrecombined cells show red fluorescence of membrane-targeted tdTomato (mT). Paraformaldehyde (PFA) fixation masks mTmG fluorescence, so in order to trace successfully recombined cells we performed immunofluorescence staining of EGFP. Immunofluorescence staining of insulin and EGFP in control mice expressing Cre recombinase (Control; +/Ins1-Cre) revealed successful labeling of β cells with EGFP. Many EGFP-stained cells were insulin-negative, revealing a loss of insulin expression in many β cells in Cnot3βKO islets (Fig. 3b, c). This was associated with a significant reduction in insulin content in Cnot3βKO islets compared with control islets (Fig. 3d). β-cell mass was reduced in Cnot3βKO mice by nearly 50% (Fig. 3e). This was associated with smaller islets, but not with a significant difference in islet number in Cnot3βKO mice, compared with control mice (Fig. 3f). These findings prompted us to determine whether decreased pancreatic insulin content and reduced β-cell mass were driven by increased β-cell death. We examined β-cell apoptosis by TUNEL assay. Cnot3βKO islets did not show extensive TUNEL staining compared with control islets (Supplementary Fig. 5).

In 8-week-old Cnot3βKO islets, many EGFP-stained cells that were insulin-negative did not stain with anti-GLUC, anti-SST, or anti-PPT antibodies, indicating that β cells did not transform into other types of islet cells (Fig. 3g). Furthermore, EGFP-stained cells in Cnot3βKO islets stained for the pan-endocrine marker, synaptophysin (SYP), indicating that these hormone-negative cells are viable and retain their endocrine phenotype (Fig. 3h).

We further examined Cnot3βKO islets using transmission electron microscopy (TEM) to detect any ultrastructural alterations in β cells. Control islets possessed β cells with typical insulin granules, manifesting characteristic electron-dense crystal cores, and with β cell–β cell contacts displaying tight junctions essential for normal insulin secretion[29]. In contrast, β cells from Cnot3βKO islets were rather heterogeneous in appearance, with few having typical insulin granules (Fig. 3i). Many were

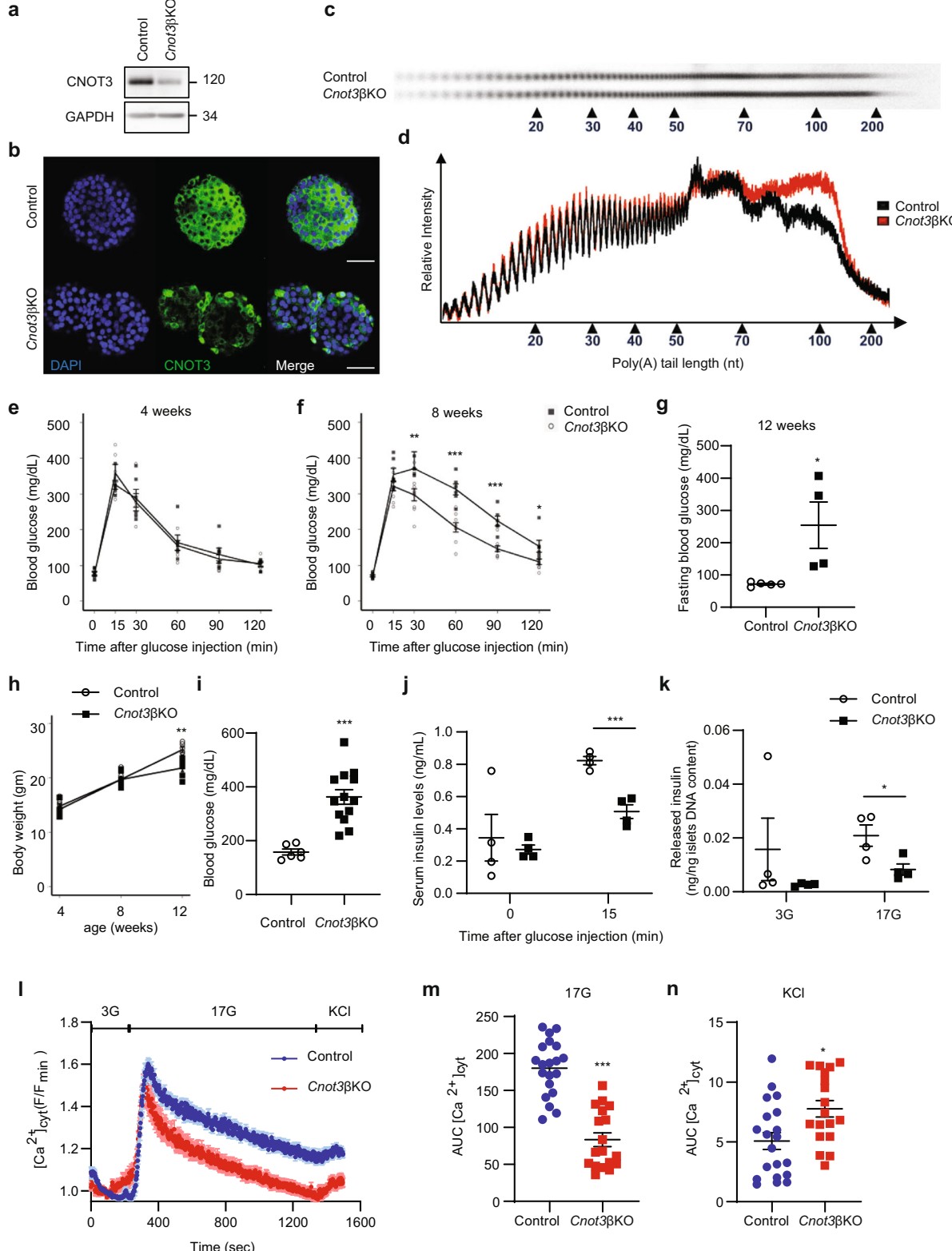

degranulated with few, or abnormally small, insulin granules (Fig. 3j, l). Also observed in *Cnot3*βKO islets were β cells with atypical rod-shaped insulin granules, or with irregular plasma membranes (Fig. 3i). We did not observe a significant difference in numbers of mitochondria in control and *Cnot3*βKO islets (Supplementary Fig. 6). The above findings suggest that diabetes in *Cnot3*βKO mice is caused by both lack of β-cell identity, and possibly by failure to achieve a fully differentiated state.

**CNOT3 is essential for β-cell maturation and identity.** Considering the above observations, we performed quantitative PCR (qPCR) to measure expression of dedifferentiation/progenitor cell markers. We observed significant upregulation of β-cell dedifferentiation/progenitor cell markers, including Ngn3, Aldh1a3, Myt1, Dcx, and Tnfrsf11b mRNAs (Fig. 4a). ALDH1A3 was recently recognized as a marker of β-cell dedifferentiation in diabetes[30,31]. Immunoblot analysis of ALDH1A3 revealed its

**Fig. 2 The loss of CNOT3 in β cells impairs glucose tolerance and causes diabetes. a** Immunoblot analysis of CNOT3 in islet lysates from 8-week-old control and *Cnot3*βKO mice. This blot is a representative of three different blots. **b** Immunofluorescence of CNOT3 (green) in islets from 8-week-old control and *Cnot3*βKO mice. Nuclei were stained with DAPI (blue). A scale bar represents 50 μm. **c** Poly(A) tail lengths of global mRNAs by bulk poly (A) assay in control and *Cnot3*βKO islets (*n* = 1, each sample was pooled from four mice). **d** Densitograms of poly(A) tail lengths shown in (**c**). Intraperitoneal glucose tolerance tests in: **e** 4- and **f** 8-week-old control and *Cnot3*βKO mice (*n* = 6–9). **g** Fasting blood glucose in 12-week-old control (*n* = 5) and *Cnot3*βKO mice (*n* = 4). **h** Body weight comparisons between control and *Cnot3*βKO mice at 4, 8 and 12 weeks of age (*n* = 6). **i** Blood glucose in 8-week-old control (*n* = 6) and *Cnot3*βKO (*n* = 13) mice fed with ND. **j** Serum insulin in 8-week-old control and *Cnot3*βKO mice after 16 h fasting (0 min) and 15 min after glucose injection (2 g/kg body weight) (*n* = 4). **k** Insulin released by islets from 8-week-old control and *Cnot3*βKO mice in response to low 3 mM glucose (3G) and high 17 mM glucose (17G) stimulation for 1 h. This experiment is a representative of three independent experiments using three different biological replicates (*n* = 3). Data are presented as mean ± SEM; *\*P* < 0.05; *\*\*P* < 0.01; *\*\*\*P* < 0.001, two-tailed Student's *t* test. **l** Cytosolic Ca$^{2+}$ ([Ca$^{2+}$] cyt) responses in control (*n* = 20) and *Cnot3*βKO (*n* = 18) islets from three 8-week-old mice from each genotype in response to low (3G), high (17G) glucose or 20 mM KCl. **m** AUC of glucose-evoked Ca$^{2+}$ traces. AUC was calculated using different baseline values for each group (0.95 for control and 1.03 for *Cnot3*βKO). **n** AUC of depolarizing stimulus, KCl, response. AUC was calculated using different baseline values for each group (1.15 for control and 0.95 for *Cnot3*βKO). Data are presented as mean ± SEM; *\*P* < 0.05; *\*\*P* < 0.01; *\*\*\*P* < 0.001, Mann–Whitney test.

significant upregulation in islets isolated from *Cnot3*βKO mice, compared with control islets (Fig. 4b, c, Supplementary Fig. 7). This suggests that β cells in *Cnot3*βKO mice exhibit a progenitor-like state.

In addition, we checked expression of a number of genes critical for β-cell function. Among mRNAs encoding the transcription factors important for β-cell function and insulin transcription that we tested (Mafa, Nkx2.2, Nkx6.1, Neurod1 and Pdx1), only Mafa mRNA was significantly reduced (Fig. 4d). Mafa, is a key β-cell transcription factor and a driver of β-cell maturation[32] that was sharply reduced at the protein level as well (Fig. 4e, Supplementary Fig. 8a). Consistent with decreased GSIS, mass spectrometry (MS) analysis revealed significant reductions in proteins involved in glucose sensing, insulin secretory granule formation, and insulin secretion (Table 1). Some affected proteins manifest no change in mRNA levels, while others were reduced, but not significantly (Fig. 4d). We observed decreased expression of glucose transporter-2 (GLUT2), in *Cnot3*βKO pancreata (Fig. 4e, Supplementary Fig. 8b). GLUT2, encoded by the *Slc2a2* gene, is a transmembrane protein involved in glucose uptake by β cells, the first step of GSIS. In support of defective maturation of CNOT3-depleted β cells, we found that urocortin 3 (Ucn3), a marker of β-cell maturation, which is also important in GSIS[33], was unchanged at the mRNA level, but was strongly decreased at the protein level (Fig. 4d, Table 1). Moreover, *Cnot3*βKO mice displayed a post-transcriptional defect in insulin biosynthesis where *Cnot3*βKO islets displayed significantly decreased expression of mRNAs encoding proteins involved in insulin biosynthesis, such as T2D-associated zinc transporter Slc30a8, proinsulin-to-insulin convertase (Pcsk2), and carboxypeptidase (Cpe)[34,35] (Fig. 4d and Table 1). These proteins have established roles in insulin processing and/or maturation of insulin secretory granules; thus, these data reveal loss of β-cell maturation markers and defects in insulin biosynthesis.

In normal mature β cells, insulin secretion is regulated by coupling glucose metabolism to insulin secretion[36]. Multiple genes involved in glycolysis are upregulated in diabetes and interfere with normal insulin secretion[37]. This prompted us to investigate glycolytic enzyme mRNAs (Ldha, Hk1, and Hk2) with other mRNAs that are normally repressed in mature β cells (Slc16a1/Mct1, Rest and Pdgfra). Loss of their repression may result in impaired β-cell function[38]. We observed significant upregulation of Slc16a1, Ldha, and Hk2 mRNAs that belong to the so-called β-cell-disallowed genes. Hk1, Rest, and Pdgfra mRNAs were upregulated, but the increases were not statistically significant (Fig. 4f). We confirmed the upregulation of MCT1, LDHA, and HK1 at the protein level as well (Fig. 4g, h, Supplementary Fig. 9, Table 1). These data support a metabolic phenotype resembling that of immature/fetal β cells.

**Transcriptome- and proteome-wide changes in *Cnot3*βKO islets**. To understand how loss of *Cnot3* affects global gene expression in β cells, we performed massive parallel RNA sequencing (RNA-seq) of control and *Cnot3*βKO islets. Comparison of gene expression profiles between control and *Cnot3*βKO islets revealed 3847 differentially expressed (DE) genes (adjusted *P* < 0.05), of which 2234 genes were upregulated and 1613 genes were downregulated (Fig. 5a). To define the biological processes affected by *Cnot3* depletion, we performed gene ontology (GO) analysis of DE genes using the DAVID gene annotation tool (Fig. 5b). Consistent with the diabetic phenotype, GO analysis showed that genes upregulated after *Cnot3* depletion are highly enriched for those involved in "immune system process", "lipid metabolic processes", and "oxidation/reduction processes", while downregulated genes are highly enriched for those involved in "regulation of insulin secretion" and "vesicle mediated transport." To gain more insight into the effect of *Cnot3* depletion on the pattern of gene expression in β cells, we applied a stricter threshold by comparing the number of significantly upregulated to downregulated genes that were DE by more than twofold. Consistent with the molecular function of CNOT3 as a positive modulator of CCR4–NOT-mediated mRNA decay, most genes that exhibited more than twofold differential expression were upregulated (Fig. 5c). We next examined whether changes in *Cnot3*βKO islet transcriptomes are reflected in the corresponding proteomes. We compared transcriptomic with proteomic data from *Cnot3*βKO islets. There was a significant strong correlation between DE genes and proteins in *Cnot3*βKO islets ($R^2$ = 0.64, *P* < 2.2e$^{-16}$), suggesting that most changes in the transcriptome are mirrored in the proteome (Fig. 5d).

Steady-state mRNA abundance is mainly determined by rates of RNA transcription and decay[39]. We used transcriptomic data to infer mRNA stability in *Cnot3*βKO islets. While exonic read counts in RNA-seq data correspond to steady-state mRNA abundance, changes in abundance of intronic reads can be used to estimate changes in transcription rate. We used this approach to infer global mRNA stability from RNA-seq data. We estimated the change in mRNA half-life as the difference of the logarithm of the fold change in exonic reads and the logarithm of the fold change in intronic reads (Δexon–Δintron)[40]. Because CNOT3 is involved in mRNA deadenylation and subsequent mRNA decay, we considered a transcript to be a target of CNOT3 if that transcript was upregulated and stabilized, as inferred from RNA-seq data. We found that 80% of mRNAs (254 of 317) that show increased Δexon–Δintron in *Cnot3*βKO compared with control islets were among the significantly upregulated mRNAs, suggesting that post-transcriptional mechanisms, including mRNA stabilization, effectively contributed to an increase of mRNAs in *Cnot3*βKO islets. In contrast, decreased intron reads in

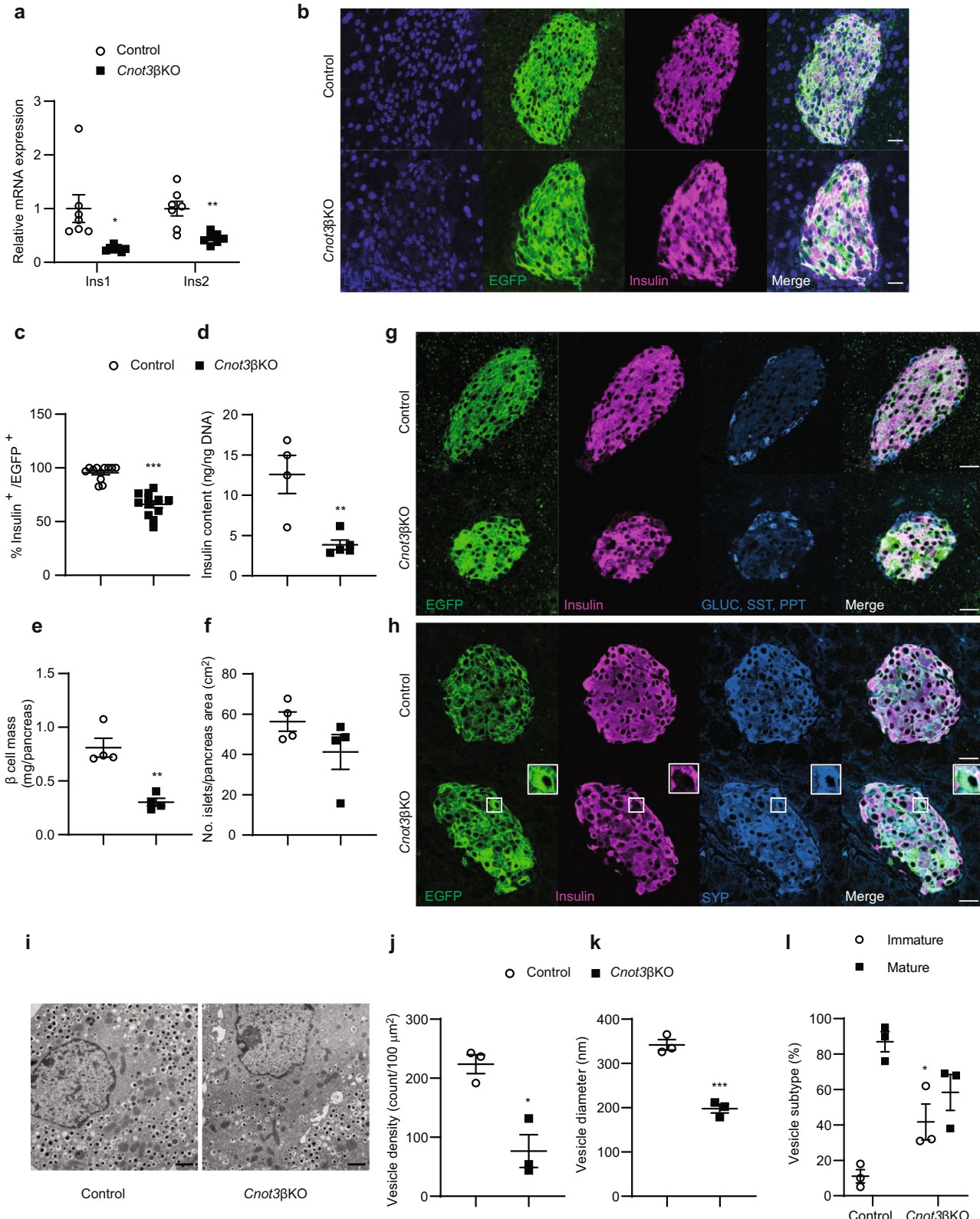

*Cnot3*βKO islets were not relevant to any significant increase of 63 mRNAs, despite increased Δexon–Δintron (Fig. 5e). We compared upregulated, stabilized mRNAs in *Cnot3*βKO islets to published RNA-seq data from *db/db* islets[41] and found 73 upregulated, stabilized mRNAs to be upregulated in *db/db* islets, suggesting that CNOT3 is involved in suppressing diabetes-related genes (Supplementary Table 1). Interestingly,

transcriptome analysis revealed Aldob to be the most upregulated (almost 1000-fold) stabilized mRNA. ALDOB is a glycolytic enzyme that is normally silenced in mature β cells[42,43]. It was dramatically upregulated at both the mRNA and protein levels (Fig. 5f–h, Supplementary Fig. 11a). This dramatic upregulation of Aldob mRNA is also possibly due to its increased transcription, as inferred from its increased pre-mRNA levels (Supplementary

**Fig. 3 The lack of CNOT3 in β cells reduces insulin expression. a** qPCR analysis of insulin mRNA isoforms (Ins1 and Ins2), normalized to the Gapdh mRNA level, in control and *Cnot3*βKO islets (*n* = 7). **b** Co-immunofluorescence staining of EGFP (green), insulin (magenta) and DAPI (blue) in pancreatic sections from 8-week-old control and *Cnot3*βKO mice. A scale bar represents 25 μm. Representative results from three 8-week-old mice from each genotype are shown. **c** Quantification data of immunofluorescence analysis presented in Fig. 3b. Each data point represents %EGFP⁺ in Insulin⁺ β cells in control and *Cnot3*βKO islets (*n* = 12) from three 8-week-old mice from each genotype. **d** Insulin content of islets from 8-week-old control (*n* = 4) and *Cnot3*βKO (*n* = 5) mice. **e** β-cell mass measurement in pancreatic sections from 8-week-old control and *Cnot3*βKO mice (*n* = 4). **f** Islet number per pancreas area in pancreatic sections from 8-week-old control and *Cnot3*βKO mice (*n* = 4). **g** Co-immunofluorescence staining of EGFP (green), insulin (magenta), GLUC (blue), SST (blue) and PPT (blue) in pancreatic sections from 8-week-old *mTmG* reporter: "control (*Cnot3*⁺,⁺; +/Ins1-Cre) and *Cnot3*βKO" mice. A scale bar represents 25 μm. **h** Co-immunofluorescence staining of EGFP (green), insulin (magenta) and SYP (blue) in pancreatic sections from 8-week-old *mTmG* reporter: "control (*Cnot3*⁺,⁺; +/Ins1-Cre) and *Cnot3*βKO" mice. A scale bar represents 25 μm. Representative results from three 8-week-old mice from each genotype are shown. **i–l** Transmission electron microscopy performed on islets isolated from 8-week-old control and *Cnot3*βKO mice (*n* = 3), including quantification of **j** vesicle density, **k** vesicle diameter and **l** the percentage of mature (black bar) and immature vesicles (white bar). A scale bar represents 1 μm. Data are presented as mean ± SEM; *P < 0.05; **P < 0.01; ***P < 0.001, two-tailed Student's *t* test.

Fig. 11b). In addition, the fructose/glucose transporter, Slc5a10, and Wnt5b were among the most stabilized mRNAs, and they were upregulated (Fig. 5f).

**Increase of β cell-disallowed genes in *Cnot3*βKO islets.** β cell-disallowed genes are a group of genes that are normally abundantly expressed in most tissues, but selectively repressed in β cells[21,44]. Two previous studies identified 68 of such genes using microarray analysis[45,46]. More recently, a novel study identified a new list of β-cell-disallowed genes, based on RNA-seq data of islets, β cells, and α cells[19]. We performed gene set enrichment analysis (GSEA) of 61 β-cell-disallowed genes identified by the latter study, which revealed enrichment of these β-cell-disallowed genes in our transcriptome dataset (Fig. 6a, b).

We inferred the mRNA stability of some of these disallowed genes together with *Aldob*, *Slc5a10*, and *Wnt5b* from transcriptome analysis (Fig. 5e). Some mRNAs were filtered out in the statistical analysis, due to the low expression values that hindered assessment of their stability. We validated this stability analysis by treatment of control and *Cnot3*βKO islets with Act D and determined their relative abundances at 4-h intervals. Aldob, Wnt5b, Cat, and Abtb2 mRNAs were significantly stabilized (Fig. 6c). On the other hand, other upregulated mRNAs: Hk1, Rest, and Yap1 were not stabilized, indicating that some mRNAs increased independent of mRNA stabilization (Fig. 6c). We could not assess the stability of Slc5a10, Slc16a1, Ldha, and Pdgfra experimentally. Yet, based on statistical evaluation of exon and intron counts, we found that among these mRNAs, Slc5a10 (logFC = 4.76, FDR = 1.7e⁻¹⁷) was stabilized.

To exclude the possibility that the observed immature phenotype is merely caused by hyperglycemia, we performed experiments on euglycemic 4-week-old *Cnot3*βKO mice. We observed significant downregulation of Mafa, Ins1, and Ins2 mRNAs, and significant upregulation of Aldob, Slc5a10, Wnt5b, Hk1, and Tnfrsf11b mRNAs, suggesting that the molecular defect results directly from the loss of *Cnot3*, rather than to hyperglycemia (Supplementary Fig. 12).

To further understand the mechanism by which CNOT3 alters mRNA stability, we asked whether affected genes interact with CNOT3. To answer this question, we carried out RNA immunoprecipitation (RIP) using MIN6 cell line lysates with an antibody specific for CNOT3 and negative control immunoglobulin G (IgG). RNA from total extracts (Input), IgG, and CNOT3-immunoprecipitated samples were analyzed by qPCR. Among β cell-disallowed genes analyzed, Slc16a1, Ldha, Cat, Abtb2, Fgf1, Igfbp4, Acot7, and Cxcl12 mRNAs were significantly enriched in CNOT3-immunoprecipitated samples, compared with IgG immunoprecipitated samples (Fig. 6d). These data suggest that the above genes interact with CNOT3. Interestingly, Yap1 is a β cell-disallowed gene that did not interact with CNOT3

(Fig. 6d) and its mRNA decay rate was not affected in *Cnot3*βKO mice (Fig. 6c). This observation implies that Yap1 upregulation after *Cnot3* deletion is an indirect effect (possibly on transcription) rather than a result of reduced decay. In summary, our data suggest that *Cnot3* depletion resulted in decreased mRNA decay rates and increased protein synthesis of target transcripts.

**Discussion**
The loss of β-cell identity and dedifferentiation are increasingly recognized as contributors to impaired β-cell function in T2D[2]. β-cell maturation involves increased expression of β-cell-specific genes as well as repression of genes that would interfere with normal β-cell function, so-called β-cell "disallowed" genes[3,45–47]. Involvement of post-transcriptional control in regulation of those genes will provide novel insights into β-cell functional maturation and pathogenesis of diabetes. Our findings suggested that CCR4–NOT complex-dependent mRNA deadenylation is crucial for regulation of β-cell-specific genes and disallowed genes, and consequently for β-cell maturation (Fig. 7). Consistent with this, a disruption of the CCR4–NOT complex might be pathogenic in diabetes.

The process of GSIS in mature β cells is regulated primarily by levels of extracellular glucose. Unlike immature β cells, which secrete insulin under low-glucose conditions, mature β cells secrete insulin in response to increases in blood glucose, thus maintaining normal glucose homeostasis without causing hypoglycemia[42]. Here, we show that both GSIS and glucose-stimulated cytosolic Ca²⁺ dynamics were impaired in *Cnot3*βKO mice, suggesting impaired glucose metabolism. Impaired glucose metabolism could be due to decreased GLUT2 expression and defects in maturity. Expression of glycolytic genes, such as *Hk1*, *Hk2*, and *Ldha*, is sharply downregulated during β-cell maturation[42,48]. We observed increased expression of these glycolytic genes and other markers of immature β cells (Slc16a1, Rest, and Pdgfra). Ectopic expression of *Hk1*, *Slc16a1*, and *Ldha* reportedly impairs glucose metabolism and GSIS[38,49,50]. Moreover, decreased expression of *Pdgfra* during maturation results in decreased proliferation of mature β cells, necessary for normal GSIS[51].

β-cell-disallowed genes are selectively silenced in functional β cells. Their repression, which ensures normal GSIS[44], is an element of β-cell identity, acquired during postnatal maturation. The molecular mechanisms involved in repressing these genes are largely unknown; thus, it is currently unclear whether there is a single silencing mechanism for all disallowed genes, or whether several molecular mechanisms are involved. Previous report suggested that epigenetic mechanisms such as DNA methylation are involved in repressing several disallowed genes (*Pdgfra*, *Acot7*, *Igfbp4* and *Fgf1*)[43]. Significantly, DNA methylation is generally essential for β-cell maturation and contributes to repression of

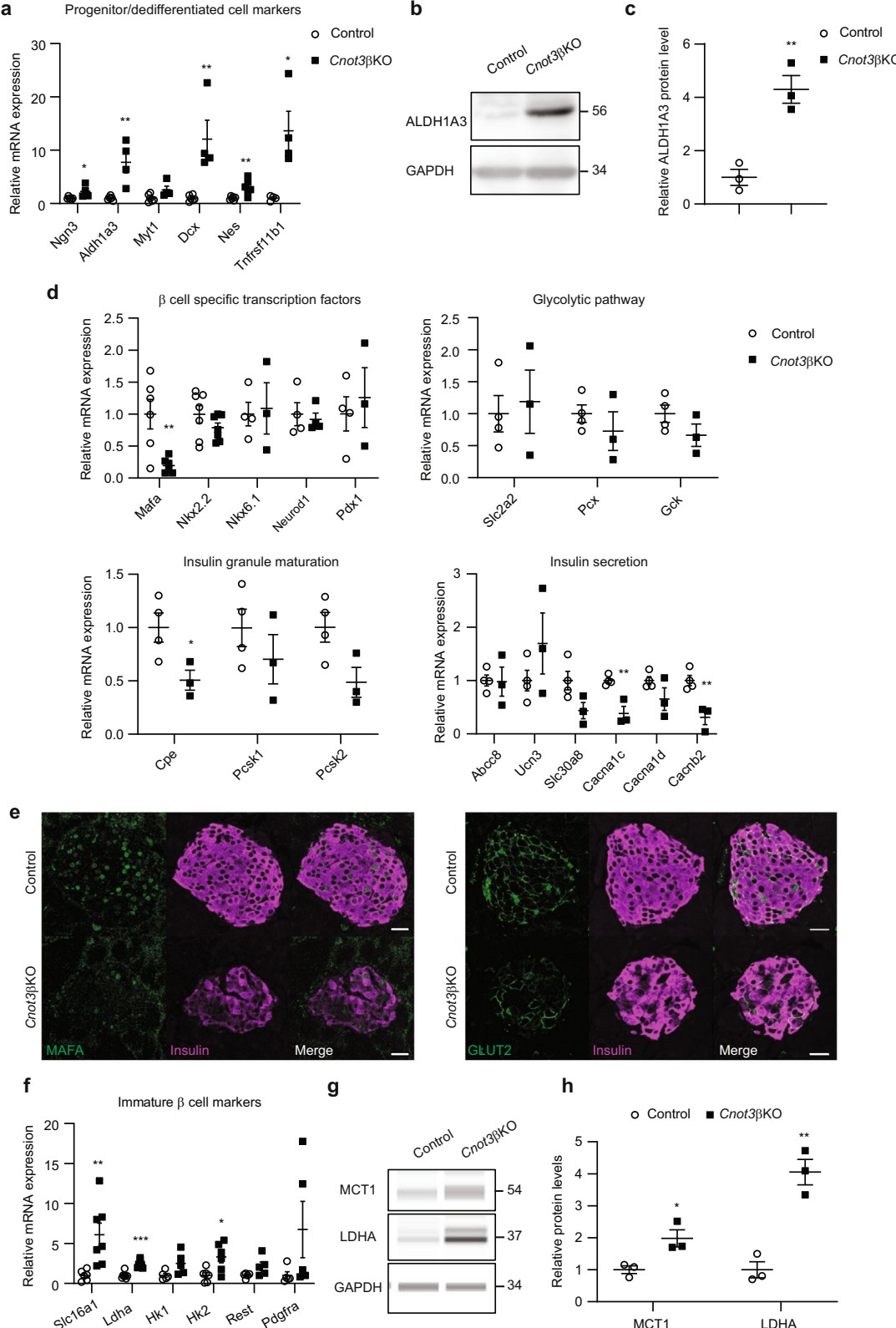

immature genes, including *Aldob*, *Hk1*, *Hk2*, and *Ldha*[42]. Histone methylation may also be involved where trimethylation of histone H3 on lysine 27 (H3K27me3), was found in the promoter regions of *Slc16a1*, *Cxcl12*, *Acot7*, *Nfib*, *Mgst1*, and *Maf*[44,52]. Importantly, miRNAs downregulate *Slc16a1*, *Acot7*, *Fcgrt*, *Igfbp4*, *Maf*, *Oat*, and *Pdgfra*, implicating post-transcriptional mechanisms in regulation of disallowed genes[7,53,54]. The CCR4–NOT complex uses

miRNAs to recognize target mRNAs[13,14]. It is conceivable that the CCR4–NOT complex is involved in miRNA-dependent regulation of β-cell-disallowed genes.

Nearly all processed eukaryotic mRNAs possess poly (A) tails at their 3′ ends. Generally, mRNAs with longer poly (A) tails are more stable than mRNAs with shorter poly (A) tails[8]. Suppression of any CCR4–NOT complex subunit in tissues results in stabilization of its

**Fig. 4 CNOT3 is essential for β-cell maturation and identity. a** qPCR analysis of progenitor cells/dedifferentiation markers, normalized to the Gapdh mRNA level, in control and Cnot3βKO islets ($n = 4$–6). **b** Immunoblot analysis of ALDH1A3 in islet lysates from 8-week-old control and Cnot3βKO mice. This blot is a representative of three different blots. **c** Band quantification of an immunoblot of ALDH1A3 ($n = 3$) in Fig. 4b. **d** qPCR analysis of β-cell-specific functional mRNAs expression categorized as β-cell-specific transcription factors, glycolytic pathway, insulin granule maturation, and insulin secretion mRNAs, normalized to the Gapdh mRNA level, in control and Cnot3βKO islets ($n = 3$–7). **e** Co-immunofluorescence staining of MAFA (green), GLUT2 (green), and insulin (magenta) in pancreatic sections from 8-week-old control and Cnot3βKO mice. A scale bar represents 25 μm. Representative results from four 8-week-old mice from each genotype are shown. **f** qPCR analysis of immature β-cell markers, normalized to the Gapdh mRNA level, in control and Cnot3βKO islets ($n = 5$–7). **g** Immunoblot analysis of MCT1 and LDHA in islet lysates from 8-week-old control and Cnot3βKO mice. This blot is a representative of three different blots. **h** Band quantification of immunoblot of MCT1 and LDHA ($n = 3$). Data are presented as mean ± SEM; *$P < 0.05$; **$P < 0.01$; ***$P < 0.001$, two-tailed Student's $t$ test.

**Table 1 Categories of proteins affected by Cnot3 KO in β cells obtained from MS analysis of Cnot3βKO islets.**

| Category | Protein symbol |
|---|---|
| Glycolysis | GCK (FC = 1, $P = 0.4$), G6PC2 (FC = 1.6, $P = 0.13$) |
| Insulin granule feature | **VAMP2** (FC = 0.6, $P = 0.01$), **SLC30A8** (FC = 0.4, $P = 0.049$), SYTL4 (FC = 0.9, $P = 0.4$), **PC1** (FC = 0.48, $P = 0.09$), **PC2** (FC = 0.58, $P = 0.007$), **CPE** (FC = 0.3, $P = 0.002$) |
| Glucose sensing and Insulin secretion | **GLUT2** (FC = 0.4, $P = 0.1$), **GLP1R** (FC = 0.58, $P = 0.0007$), **UCN3** (FC = 0.4, $P = 0.02$) |
| β cell disallowed | **ACOT7** (FC = 1.55, $P = 0.1$), **HK1** (FC = 2, $P = 0.003$), **ALDOC** (FC = 3.1, $P = 0.02$), **LDHA** (FC = 1.89, $P = 0.04$) |

Affected β-cell markers are in bold.

target mRNAs, due to elongated poly (A) tails[24,25,55]. In our study, insufficient deadenylation observed in Cnot3βKO islets (Fig. 2c, d), is coincident with upregulation and stabilization of mRNAs, including Aldob, Slc5a10, and Wnt5b (Fig. 5f). Increased expression of Aldob and Slc5a10 is associated with decreased insulin secretion[37,56], suggesting their contribution to impaired GSIS. Importantly, upregulation of Aldob, Slc5a10, and Wnt5b mRNAs is correlated with diabetes[37,56–58]. These results agree with our previous findings that the CCR4–NOT complex is involved in regulation of energy metabolism genes[15–18]. This study provides evidence that the CCR4–NOT complex is involved in repressing Aldob, Slc5a10, Wnt5b, and several disallowed genes through both direct and indirect mechanisms. We showed that Cnot3 KO results in upregulation and increased stability of Aldob, Slc5a10, Wnt5b, Cat, and Abtb2 mRNAs (Fig. 6c). The finding that CNOT3 directly interacts with Wnt5b, Cat, and Abtb2 mRNAs (Fig. 6d) suggests direct repression of these genes by the CCR4–NOT complex.

On the other hand, suppression of CNOT3 is likely to have indirect effects on gene expression partly through expression changes in key transcription factors. We and others have recently reported that Mafa, Pax6, Rfx6, and Nkx2.2 act as repressors of several disallowed genes, and changes in any of these factors may be involved in upregulation of disallowed genes[32,59–61]. Among these repressors, Mafa was significantly decreased in Cnot3βKO islets (Fig. 4d, e). Therefore, it is possible that decreased Mafa in Cnot3βKO islets contributes to increased expression of disallowed genes. Downregulation of Mafa could also explain upregulation of endocrine progenitor cell markers observed in Cnot3βKO. Upregulation of these progenitor markers, as a result of Mafa downregulation, indicates a failure to attain a fully differentiated state in β cells and upregulates β-cell-disallowed genes[32]. Indeed, Slc16a1 was upregulated, but not stabilized. Thus, we propose that Slc16a1 is upregulated indirectly by CNOT3 through Mafa repression. This agrees with its upregulation observed in dedifferentiated β cells of Mafa KO mice[32]. The Mafa suppression mechanism was not determined in this study. Taken together, our data and those of others suggest that repression of disallowed genes is achieved by multiple mechanisms.

Our findings raise the possibility that disruption of the CCR4–NOT complex could be involved in the loss of β-cell identity

in diabetes, possibly through de-repression of β-cell immature and disallowed genes. We observed decreased expression of CNOT3 in db/db mice islets. Moreover, glucotoxicity and lipotoxicity significantly reduced CNOT3 levels in MIN6 cells. This agrees with proteomic analysis of human islets that revealed significant reduction of CNOT3 during high-glucose treatment[62]. In addition, enrichment of immune-related mRNAs among the upregulated mRNAs in Cnot3βKO islets suggests an involvement of enhanced immune response in impaired β cells. It should be noted that enhanced immune response is observed in mouse tissues lacking CCR4–NOT complex function, such as liver and adipose tissues[16,18,55,63]. In our model, we cannot determine whether the increased immune response in Cnot3βKO mice is a cause or a consequence of β-cell dysfunction. Nevertheless, our findings suggest possible relationship between CCR4–NOT complex dysfunction and diabetes (both T1D and T2D). This requires further study including participation of human diabetes patients.

Several lines of evidence have shown that the loss of CCR4–NOT complex subunits results in serious abnormalities in embryonic development and tissue function[10]. Importantly, the CCR4–NOT complex contributes to maintenance of both pluripotency of embryonic stem cells and tissue function/identity via mRNA deadenylation. The CCR4–NOT complex promotes the decay of mRNAs that are relevant to immature tissue state, cell death, and proliferation to ensure differentiation or normal tissue development[16,24,55]. In contrast, the CCR4–NOT complex decreases mRNAs responsible for differentiation to maintain pluripotency[64–66]. Therefore, the CCR4–NOT complex contributes to proper mRNA expression by decreasing unnecessary mRNAs depending on the context. The difference in mRNA-binding proteins that recruit the CCR4–NOT complex to target mRNAs could explain its diverse functions. It is also possible that tissue-specific RBPs compete with the interaction of the CCR4–NOT complex with particular mRNAs, as in the case of HuR and miRNA-induced silencing complex[67]. Global understanding of target mRNA recognition of the CCR4–NOT complex depending on the context will help to delineate the molecular basis of tissue-specific CCR4–NOT complex function.

In conclusion, our work identifies CNOT3 as an important post-transcriptional regulator of β-cell identity and function,

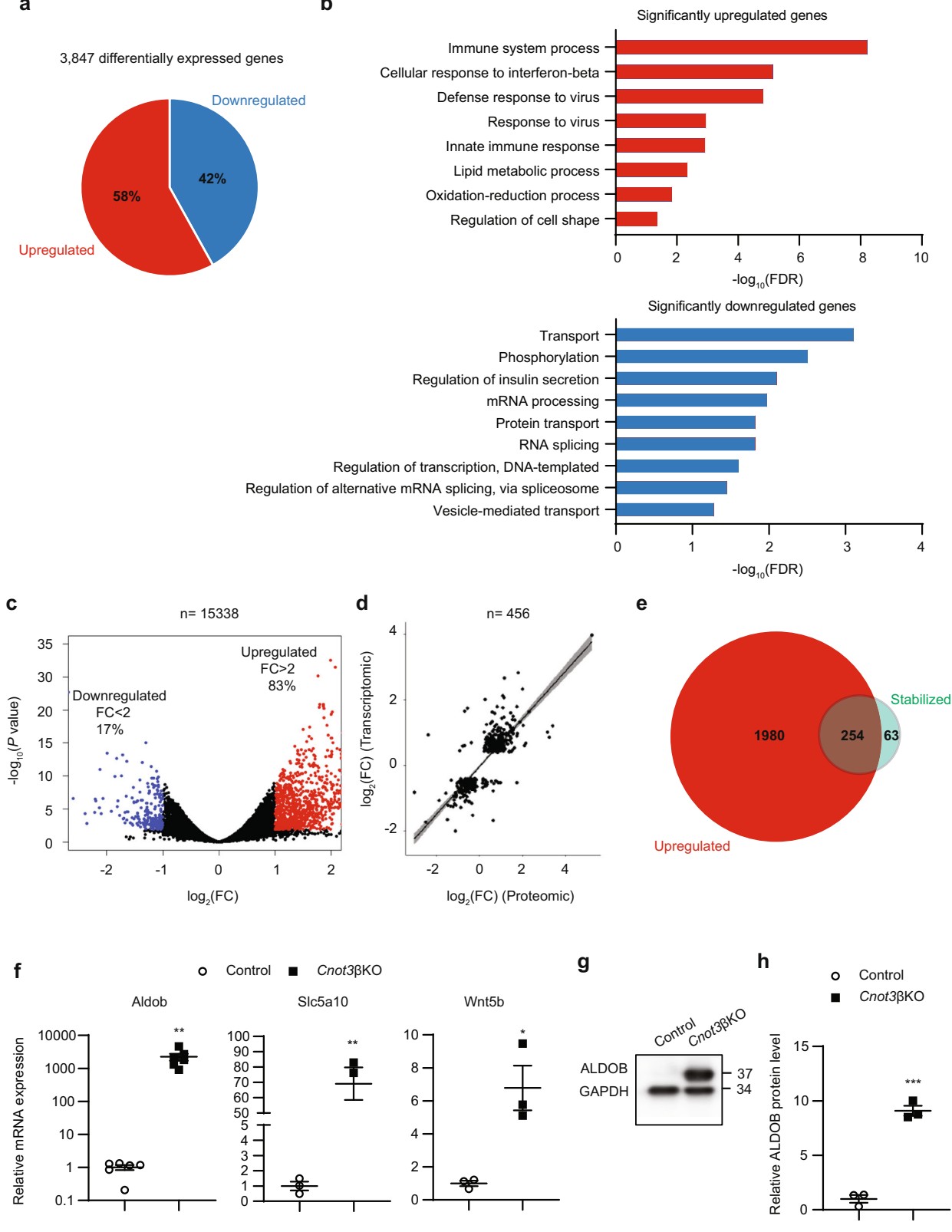

and raises the possibility that alterations in CNOT3 expression and/or dysregulation of the CCR4–NOT complex contribute to β-cell failure in diabetes. It further raises the possibility that CNOT3 might offer a promising therapeutic target to protect β-cell function or to generate β cells for treatment of diabetes.

## Methods

**Mice**. To suppress *Cnot3* expression in β cells, we crossed mice carrying *Cnot3* floxed alleles (*Cnot3^flox/flox*)[16,26] with *Ins1-Cre* mice (Riken Bioresource Center, # RBRC09525), in which Cre recombinase is knocked in at the *Ins1* locus[68]. Recombination is expected to be restricted largely to β cells, and to be minimal at other sites, including the brain, where *Ins1* is not expressed[69]. *mTmG* reporter mice (Jackson Laboratory, # 007676) were used to trace β cells in which *Ins1-Cre-*

**Fig. 5 Global gene expression changes in *Cnot3*βKO islets. a** 3847 genes were differentially expressed (DE) in islets following CNOT3 depletion in β cells. The graph indicates the percentage distribution of genes differentially expressed ($P < 0.05$) in *Cnot3*βKO compared with control islets ($n = 3$, each sample was pooled from two mice). Multidimensional scaling (MDS) plot is presented in Supplementary Fig. 10. **b** GO analysis of mRNAs significantly upregulated (red) and downregulated (blue) in *Cnot3*βKO islets. Bar charts of GO terms (Biological Process) ranked by FDR (<0.05) are shown. Gene lists included in each GO term are summarized in Supplementary Data 1. **c** Volcano plot of DE genes indicating the percentage distribution of DE ($P < 0.05$) genes by more than twofold upregulation (red) or downregulation (blue). **d** Pearson correlation analysis between differentially expressed genes and proteins identified by RNA-seq analysis and MS analysis of control and *Cnot3*βKO islets. **e** Venn diagram showing the overlap of significantly upregulated (red) and stabilized (green) mRNAs in *Cnot3*βKO compared with control islets. **f** qPCR analysis of three top upregulated and stabilized mRNAs (Aldob, Slc5a10 and Wnt5b), normalized to the Gapdh mRNA level, in control and *Cnot3*βKO islets ($n = 3$–6). **g** Immunoblot analysis of ALDOB in islet lysates from 8-week-old control and *Cnot3*βKO mice ($n = 3$). This blot is a representative of three different blots. **h** Band quantification of immunoblot of ALDOB ($n = 3$). Data are presented as mean ± SEM; *$P < 0.05$; **$P < 0.01$; ***$P < 0.001$, two-tailed Student's *t* test.

mediated recombination was induced. Primers used for genotyping are listed in Supplementary Table 2. β-cell-specific *Cnot3* KO mice (*Cnot3*βKO) were used for experiments, and their littermates (*Cnot3*flox/flox) were used as controls, unless otherwise stated (*Ins1-Cre* mice were used as controls for *Cnot3*βKO expressing *mTmG* reporter gene). All experiments were performed on 8–10-week-old male mice, unless stated otherwise. *db/db* mice were used as a model of T2D[22] and compared with +/*db* mice, purchased from CLEA Japan. We maintained mice on a 12-h light/dark cycle in a temperature-controlled (22 °C) barrier facility with free access to water and either a normal chow diet (NCD, CA-1, CLEA Japan) or a HFD (HFD32, CLEA Japan). All mouse experiments were approved by the Animal Care and Use Committee of Okinawa Institute of Science and Technology (OIST) Graduate University, Okinawa, Japan.

**Islet isolation**. Pancreata were perfused with Collagenase P (Sigma) by injection in the common hepatic bile duct at a concentration of 1 mg/mL in Hank's buffered salt solution (HBSS) medium (Invitrogen) supplemented with 1% bovine serum albumin, BSA (Roche). Pancreata were then removed and dissociated in a 37 °C water bath for 16 min. Dissociated pancreata were passed through a metal sieve to remove undigested tissue chunks, followed by washing in HBSS supplemented with 0.1% BSA. Islets were then separated onto a gradient of Histopaque 1077 (Sigma) overlaid with RPMI 1640 medium (Gibco) supplemented with 10% fetal bovine serum (FBS) and 1% penicillin–streptomycin (10,000 U/mL, Gibco). Afterward, islets were collected from the boundary of Histopaque 1077 and RPMI 1640 and passed through a 70-μm cell strainer into a cell culture dish with RPMI 1640 supplemented with 10% FBS and 1% penicillin/streptomycin, from which islets were hand-picked for experiments to avoid contamination with exocrine tissue.

**Cell lines**. MIN6 (mouse insulinoma-6) β-cell line, a well-established model for the study of β-cell function, was used for some in vitro investigations[70]. MIN6 cells were obtained from Dr. Susumu Seino, Kobe University Graduate School of medicine. MIN6 cells were cultured in high-glucose Dulbecco's modified Eagle's medium (Wako) supplemented with 10% FBS and 1% penicillin–streptomycin. For induction of glucotoxic stress, the glucose concentration was increased to 50 mM by dissolving glucose in culture medium followed by sterile filtration.

**Fatty acid solution preparation and MIN6 cell line treatment**. Palmitate stock solution was prepared by dissolving palmitate (Sigma P9767) in 50% ethanol at 70 °C to a final concentration of 100 mM. The stock solution was then diluted in culture medium with 0.5% BSA to a final concentration of 0.5 mM. Palmitate was allowed to complex with BSA for 30 min at 37 °C before being added to the cells. Cells not treated with palmitate were treated in the same manner, but palmitate stock solution was replaced with vehicle (50% ethanol).

**Immunoblot analysis**. Pancreatic islets and MIN6 cells were lysed using TNE lysis buffer (1% NP-40, 50 mM Tris–HCl [pH 7.5], 150 mM NaCl, 1 mM EDTA, 1 mM phenylmethylsulfonylfluoride, 10 mM NaF). All immunoblot analyses were done following SDS-PAGE standard protocols except for MCT1 and LDHA. We could not detect them by conventional western blotting and used automated Simple Wes system (ProteinSimple) for their detection.

We used an automated Simple Wes system, according to the manufacturer's instructions with a 12–230 kDa Separation Module (ProteinSimple SM-W004) and the Anti-Rabbit Detection Module. Four microliters of islet lysate was combined with 1 μL Fluorescent Master Mix and heated at 95 °C for 5 min. The biotinylated ladder, samples, antibody diluent, primary antibodies (in antibody diluent), HRP-conjugated secondary antibodies, and chemiluminescent substrate were pipetted into the plate (part of the Separation Module). Instrument default settings were used: stacking and separation at 375 V for 25 min; blocking reagent for 5 min; primary and secondary antibody both for 30 min; Luminol/peroxide chemiluminescence detection for ~15 min. Details of primary and secondary antibodies used are listed in Supplementary Tables 3 and 4, respectively.

**Glucose tolerance tests**. Mice were fasted for 16 h, followed by fasting blood glucose measurement using blood samples taken from the tail vein. Mice were then injected intraperitoneally with glucose (2 g/kg body weight) and blood glucose was measured in tail vein blood samples at 15, 30, 60, and 90 min using the Glutest Pro glucometer (Sanwa Kagaku Kenkyusho, Japan).

**GSIS assay in vivo**. Mice were fasted for 16 h, followed by blood sample collection from a facial vein to evaluate serum insulin levels under no glucose stimulation. Mice were then injected intraperitoneally with glucose (2 g/kg body weight) and blood samples were collected from the facial vein 15 min post-glucose injection. Blood samples were allowed to clot in vacutainer tubes (BD Japan) at room temperature followed by centrifugation for 10 min at 4000 rpm and separation of serum. Serum insulin levels were measured using an ultrasensitive mouse insulin ELISA kit (Takara).

**GSIS assay on islets ex vivo**. Ten islets were pre-cultured in 450 μL Krebs Ringer buffer (KRB) (140 mM NaCl, 3.6 mM KCl, 0.5 mM NaH$_2$PO$_4$, 2 mM NaHCO$_3$, 1.5 mM CaCl$_2$, 0.5 mM MgSO$_4$, 10 mM HEPES, 0.25% BSA), pH = 7.4, containing 3 mM glucose for 1 h in open Eppendorf tubes. Supernatants were discarded and replaced with 450 μL KRB containing either 3 mM or 17 mM glucose for 1 h. Supernatants were collected after 1 h. Finally, islets were treated with 1 mL 1.5% HCl in ethanol and homogenized by sonication three times (30 s on/off). Acid/ethanol lysates were then centrifuged. Supernatants were collected to measure insulin content and residual islet fragments were used for DNA extraction. Insulin in KRB supernatant (released insulin) and islet lysates (islet insulin content) was measured using an ultrasensitive mouse insulin ELISA kit (Takara). Both released insulin and islets insulin content were normalized to islet DNA content.

**Measurement of intracellular free calcium (Ca$^{2+}$)**. Functional multicellular Ca$^{2+}$-imaging was performed on isolated islets previously incubated with Cal-520 acetoxymethyl (AM; 2 μM; Stratech) for 45 min at 37 °C in KRB supplemented with 3 mM glucose. Fluorescent imaging was performed using a Zeiss Axiovert microscope equipped with a 10–20×/0.3–0.5 NA objective, a Hamamatsu image-EM camera coupled to a Nipkow spinning disk head (Yokogawa CSU-10). Volocity software (PerkinElmer Life Sciences) provided the interface. Islets were kept at 37 °C and constantly perfused with KRB-containing 3 mM, 17 mM glucose or 20 mM KCl. For each experiment, ~20 islets were used, isolated from 3 animals per genotype. Imaging data were analyzed with ImageJ software using an in-house macro (available upon request). Fluorescent traces ($F$) were normalized to baseline recorded under 3 mM glucose ($F_{min}$). The AUC at 17 mM glucose was calculated using different fluorescence baseline values (prior perfusion with high glucose) for each genotype group. The same approach was adopted to measure the AUC during KCl imaging, where baseline values prior to stimulation with KCl were used.

**Pearson (R)-based connectivity and correlation analyses**. Correlation analyses between the Ca$^{2+}$ signal time series for all cell pairs of an imaged islet were performed in MATLAB using a modified custom-made script from the one previously described[27]. Briefly, a noise reduction function (effectively a rolling average) was applied to smooth noisy data and all traces were normalized to minimum (basal) fluorescence ($F_{min}$). The correlation function $R$ between all possible (smoothed) cell-pair combinations (excluding the autocorrelation) was assessed using the Pearson's correlation. Data are displayed as heatmap matrices, indicating individual cell-pair connections on each axis (min. = 0; max. = 1). Cartesian coordinates of imaged cells were then taken into account, while constructing connectivity line maps. Cell pairs were connected with a straight line, the color of which represented the correlation strength, based on a color-coded light–dark ramp ($R = 0.1$–0.25 [blue], $R = 0.26$–0.5 [green], $R = 0.51$–0.75 [yellow], $R = 0.76$–1.0 [red]). Positive $R$ values (excluding auto-correlated cells) and the percentage of cells that were significantly connected to one another were averaged and compared between groups.

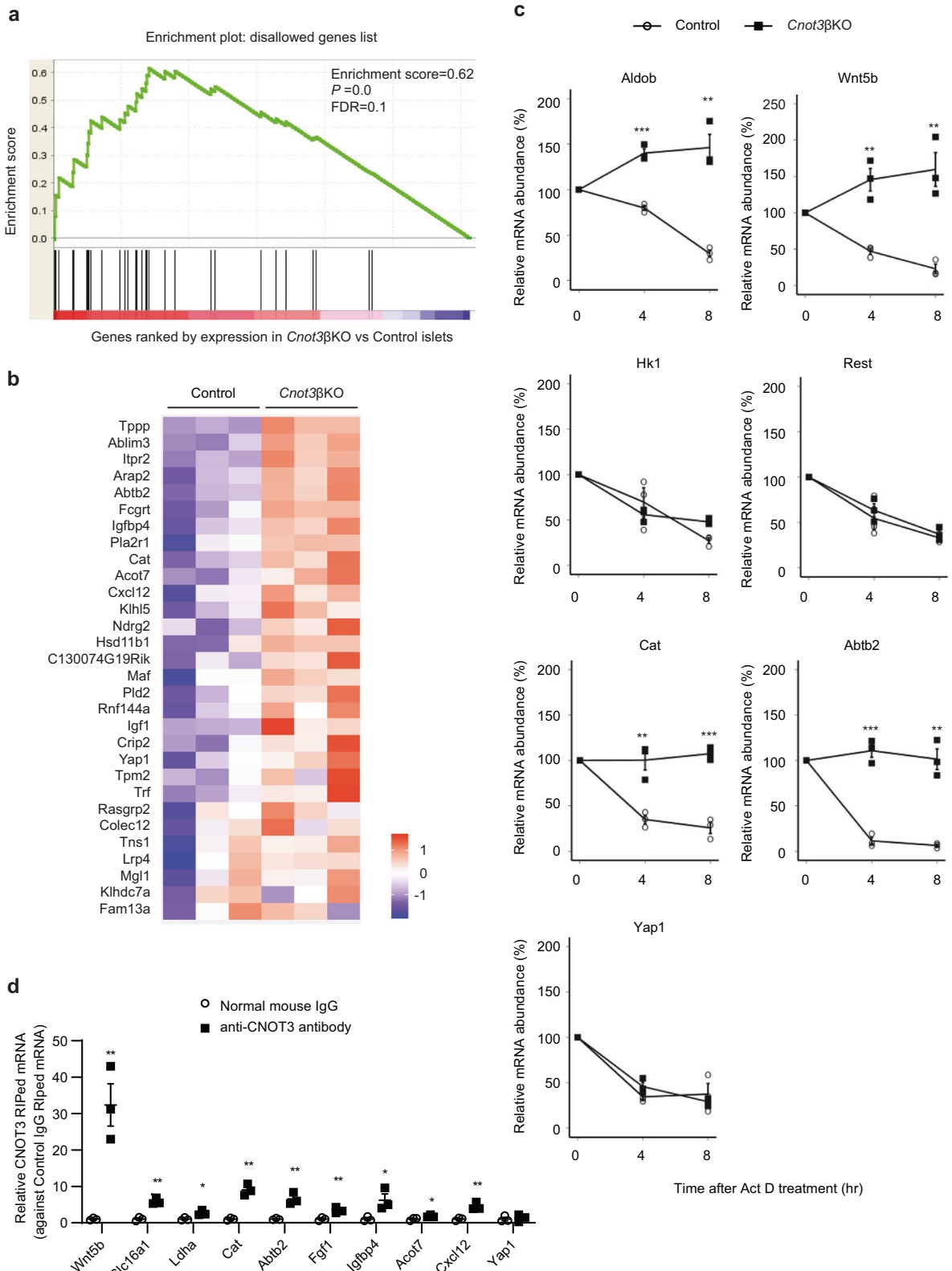

**Fig. 6 Derepression of β cell-disallowed genes and their mRNA stabilization in *Cnot3*βKO islets. a** GSEA of β-cell-disallowed genes, identified in Pullen et al.[19], in *Cnot3*βKO islets. **b** Heatmap of 30 disallowed genes showing mRNA expression obtained from RNA-seq analysis of control and *Cnot3*βKO islets. **c** Decay curves of the indicated mRNAs. Total RNA was prepared from control and *Cnot3*βKO islets treated with Act D for 0, 4, or 8 h. Relative mRNA levels were determined by qPCR and normalized to the Gapdh mRNA level. mRNA levels without Act D treatment (0 h) were set to 100% (*n* = 3). **d** RIP-qPCR in MIN6 cells using normal anti-mouse and anti-CNOT3 antibodies revealing the interaction of CNOT3 with the indicated mRNAs that adversely affect β cell function (*n* = 3). Data are presented as mean ± SEM; *P < 0.05; **P < 0.01; ***P < 0.001, two-tailed Student's *t* test.

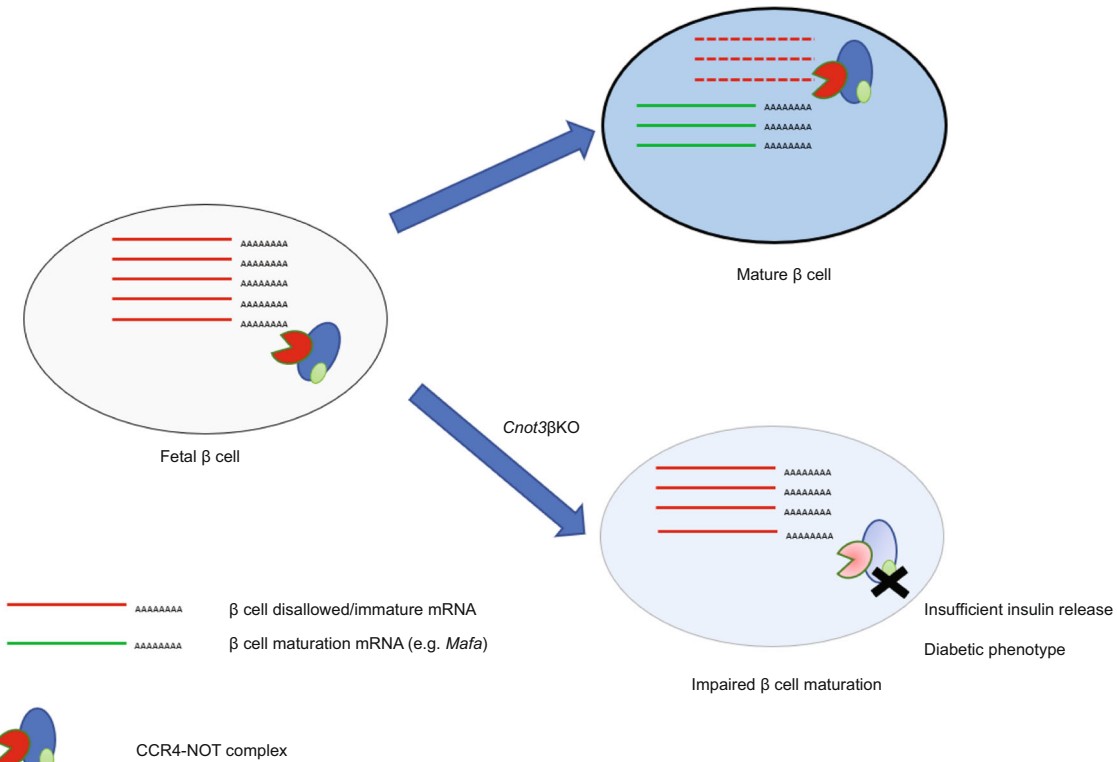

**Fig. 7 A schematic model of role of CCR4–NOT complex-mediated deadenylation in β-cell maturation.** CNOT3, one key subunit of the CCR4–NOT complex, is required for proper function of the CCR4–NOT complex. CNOT3 (Green oval) regulates expression of some β-cell-disallowed/immature genes, possibly through CCR4–NOT complex-mediated mRNA decay. Therefore, *Cnot3* KO in β cells impairs β-cell maturation due to failure to repress β-cell-disallowed/immature genes.

**Immunohistochemistry and immunofluorescence**. All immunohistochemistry was performed on pancreatic sections from control and *Cnot3*βKO mice, except for CNOT3 immunofluorescence staining, which was performed on control and *Cnot3*βKO islets.

*Pancreatic section preparation and Immunohistochemistry*: Pancreata were fixed overnight in 4% PFA phosphate-buffered solution (Wako) at room temperature. They were then transferred to 70% ethanol and subsequently embedded in paraffin and sectioned at 3 μm. Some sections were stained with H&E. Other sections were used for immunofluorescence. All sections for immunofluorescence underwent dewaxing with xylene and rehydration with ethanol (100%, 95%, 70%, and 50%) by incubation for 3 min at room temperature. Antigen retrieval was performed using 10 mM sodium citrate buffer, pH = 6, at 120 °C in a pressure cooker for 10 min. Once sections cooled to room temperature, tissue samples were blocked with blocking buffer (2% BSA solution in phosphate-buffered saline (PBS) and 0.05% Tween) for 30 min to 1 h at room temperature. Samples were then incubated with primary antibodies overnight at 4 °C. On the following day, samples were incubated with secondary antibodies for 1–2 h in the dark at room temperature. A TUNEL assay was performed using an In Situ Cell Death Detection Kit, TMR red (Roche), following the manufacturer's protocol.

*Immunofluorescence staining of CNOT3*: Islets were fixed in 4% PFA phosphate-buffered solution (Wako) for 10 min at room temperature. Then samples were washed twice in PBS for 5 min. Residual PFA was quenched by incubation with 100 mM glycine for 10 min at room temperature, followed by two washes with PBS. Islets were then permeabilized in 0.5% Triton-X-100 for 5 min at room temperature and rinsed twice with PBS. Islets were blocked with blocking buffer and incubated with primary and secondary antibodies in the same manner as described for pancreatic sections. Finally, samples were incubated with Ultra Cruz hard-set mounting medium with DAPI (Santa Cruz). Details of primary and secondary antibodies used are listed in Supplementary Tables 5 and 6, respectively.

**Morphometric image analysis**. All immunohistochemistry/immunofluorescence images were acquired using a Leica confocal microscope (TCS SP8 Leica). All images were processed using Fiji (www.fiji.sc) and Imaris (v.2.9) softwares. To calculate β-cell mass, pancreatic sections were stained with insulin (β-cell area). Then, β-cell area was divided by total pancreatic area and multiplied by pancreatic weight (mg).

**Transmission electron microscopy**. We fixed islets in 2.5% glutaraldehyde in 0.1 M cacodylate buffer (pH 7.4) for 30 min at room temperature. After three washes with 0.1 M cacodylate buffer, islets were fixed in 1% cacodylate-buffered osmium tetroxide for 1 h at room temperature, then dehydrated in a graded ethanol series, and embedded in epoxy resin. Ultrathin sections (50 nm) of islets were cut with a diamond knife, placed on copper grids, and stained with 4% uranium acetate for 30 min and then with Sato's lead staining solution. Samples were examined using a JEOL JEM-1230R TEM.

**RNA extraction and quantitative PCR**. Total RNA was extracted from islets or MIN6 cells with Isogen II, according to the manufacturer's protocol (Nippon Gene). RNA purity and concentration were evaluated by spectrophotometry using a NanoDrop ND-2000 (Thermo Fisher). The quality of RNA was assessed using an Agilent 2100 Bioanalyzer microfluidics-based platform (Agilent Technologies, Inc.). To measure mRNA stability, islets were treated with Act D (5 μg/mL), a transcription inhibitor, and total RNA was extracted at the indicated time points and subjected to qPCR. Total RNA (1 μg) was used for reverse transcription with oligo(dT)12–18primer (Invitrogen) using the SuperScript III First-Strand Synthesis System (Invitrogen). qPCR reactions were carried out using TB Green Premix Ex Taq (Takara) and the ViiA 7 Real-Time PCR System (Applied Biosystems). Gapdh mRNA levels were used for normalization. Relative mRNA expression was determined by ΔΔCT method. Primers used for qPCR reactions are listed in Supplementary Table 7.

**Bulk poly(A) tail assay**. Bulk poly(A) tail assay was conducted as previously described. Total RNA (2 μg) was labeled with [5′-³²P] pCp (cytidine 3′,5′-bis [phosphate]) (0.11 pmol/μL in a total reaction volume of 30 μL) (PerkinElmer; NEG019A) using T4 RNA ligase 1 (NEB, M0204S) at 16 °C overnight. Labeled RNAs were incubated at 85 °C for 5 min and placed on ice. Then, labeled RNAs were digested with Ribonuclease A (50 ng/μL, Sigma) and Ribonuclease T1 (1.25 U/μL, Thermo Fisher Scientific) at 37 °C for 2 h in digestion buffer (100 mM Tris–HCl [pH 7.5], 3 M NaCl, 0.5 μg/mL yeast tRNA). Reactions were stopped by adding 5× stop solution (10 mg/mL Proteinase K, 0.125 M EDTA, 2.5% SDS) and subsequently incubating at 37 °C for 30 min. After adding 400 μL of RNA precipitation buffer (0.5 M NH₄OAc, 10 mM EDTA), digested RNA samples were purified by phenol–chloroform extraction and isopropanol precipitation. Final products (10 μL) were mixed with RNA Gel loading Dye (NEB, R0641) and

incubated at 95 °C for 2 min. Samples were fractionated on an 8 M urea-10% polyacrylamide denaturing gel (0.8 mm thick). Marker (Prestain Marker for small RNA Plus, BioDynamics Laboratory DM253) was also loaded. The gel was analyzed with a Typhoon FLA 9500 Fluorescence Imager (GE Healthcare). Band intensity was quantified using ImageJ.

**RNA sequencing**. RNA-seq was performed by the next-generation sequencing section at OIST Graduate University. Hundred nanograms of total RNA was used for RNA-seq library preparation with a TruSeq Stranded mRNA Library Prep Kit for NeoPrep (NP-202-1001; Illumina) that allows polyA-oligo(dT)-based purification of mRNA. The manufacturer's protocol was employed with minor modification and optimization as follows. Custom dual index adapters were ligated at the 5′ and 3′-end of libraries, and PCR was performed for 11 cycles. One hundred fifty-base-pair paired-end read RNA-seq was performed with a Hiseq 3000/4000 PE Cluster Kit (PE-410-1001; Illumina) and a Hiseq 3000/4000 SBS Kit (300 Cycles) (FC-410-1003; Illumina) on a Hiseq4000 (Illumina), according to the manufacturer's protocol.

**Bioinformatic processing of RNA-seq data**. Paired-end RNA-seq data were mapped to the *Mus musculus* reference strain mm10 UCSC using StrandNGS, next-generation sequencing analysis software. Counts for each sample were imported into the R statistical environment. Genes without an expression level of at least one read per million mapped reads in at least three samples were removed before differential-gene expression testing between control, and *Cnot3*βKO islet RNA replicates, using the edgeR function in the Bioconductor package edgeR. Genes that were DE with an FDR adjusted P value (padj) <0.05 were considered statistically significant and included in enrichment testing. We identified GO terms enriched among significantly upregulated and downregulated genes using the DAVID annotation tool (https://david.ncifcrf.gov/). GO terms were considered significantly enriched if they had an FDR value < 0.05.

**Sample preparation for proteomic analysis**. Samples were prepared for liquid chromatography/mass spectrometry (LC/MS) using the phase-transfer surfactant method, with minor modifications. First, proteins were extracted from islets and solubilized using buffer containing 12 mM sodium deoxycholate, 12 mM sodium N-dodecanoylsarcosinate, and 100 mM Tris pH 9.0, with EDTA-free Protease Inhibitor Cocktail (Roche, Switzerland). Islet samples were sonicated for 10 min using a Bioruptor (Cosmo Bio, Japan) on high power with 1-min on/1-min off cycles. Cell debris was removed after centrifugation at $18,000 \times g$ for 20 min at 4 °C. Protein concentrations were adjusted to a uniform concentration for a set of samples (0.5–1.0 µg/µL), and between 5 and 20 µg protein was used for peptide preparation. Cysteine–cysteine disulfide bonds were reduced with 10 mM dithiothreitol at 50 °C for 30 min. Free thiol groups were alkylated with 40 mM iodoacetamide in the dark at room temperature for 30 min. Alkylation reactions were quenched with 55 mM cysteine at room temperature for 10 min. Samples were diluted with 2.76 volumes of 50 mM ammonium bicarbonate. Lysyl endopeptidase (Wako, Japan) and trypsin (Promega, USA) were added at a 50:1 ratio of sample protein:enzyme (w/w) and samples were digested for 14 h at 37 °C. Afterward, 1.77 volumes ethyl acetate were added, and samples were acidified with trifluoroacetic acid (TFA), which was added to 0.46% (v/v). Following centrifugation at $12,000 \times g$ for 5 min at room temperature, samples separated into two phases. The upper organic phase containing sodium deoxycholate was removed, and the lower aqueous phase containing digested tryptic peptides was dried using a centrifugal vacuum concentrator. Digested peptides were dissolved in 300 µL of 0.1% (v/v) TFA in 3% acetonitrile (v/v) and samples were desalted using MonoSpin $C_{18}$ columns (GL Sciences Inc., Japan). Peptides were eluted from $C_{18}$ columns using 0.1% TFA in 50% acetonitrile and dried in a vacuum concentrator. Tryptic peptides were dissolved in 0.1% (v/v) formic acid in 3% (v/v) acetonitrile for MS analysis.

**MS measurement**. Samples were measured using a Q-Exactive Plus Orbitrap LC–MS/MS System (Thermo Fisher Scientific, USA), equipped with a Nanospray Flex ion source. The same amount of peptide was injected for each sample in a given set of samples, which was typically 300–600 ng in a volume of 2 to 5 µL. Peptides were separated on a 3-µm particle, 75-µm inner diameter, 12-cm filling length $C_{18}$ column (Nikkyo Technos Co., Ltd, Japan). A flow rate of 300 nL/min was used with a 2-h gradient (1–34% solvent B in 108 min, 34–95% solvent B in 2 min, with a final wash at 95% solvent B for 10 min, where solvent A was 0.1% (v/v) formic acid in LC/MS grade water and solvent B was 0.1% (v/v) formic acid in 80% (v/v) acetonitrile). The mass spectrometer ion transfer tube temperature was 250 °C and 2.0 kV spray voltage was applied during sample measurement.

For data-dependent acquisition (DDA), full MS spectra were acquired from 380 to 1500 $m/z$ at a resolution of 70,000. The AGC target was set to $3e^6$ with a maximum injection time (IT) of 100 ms. MS2 scans were recorded for the top 20 precursors at 17,500 resolution with an AGC of $1e^5$ and a maximum IT of 60 ms. The first mass was fixed at 150 $m/z$. The default charge state for the MS2 was set to 2. HCD fragmentation was set to normalized collision energy of 27%. The intensity threshold was set to $1.3e^4$, charge states 2–5 were included, and dynamic exclusion was set to 20 s.

For data-independent acquisition (DIA), data were acquired with 1 full MS and 32 overlapping isolation windows constructed covering the precursor mass range of 400–1200 m/z. For full MS, resolution was set to 70,000. The AGC target was set to $5e^6$ and maximum IT was set to 120 ms. DIA segments were acquired at 35,000 resolution with an AGC target of $3e^5$ and an automatic maximum IT. The first mass was fixed at 150 $m/z$. HCD fragmentation was set to normalized collision energy of 27%.

**Protein identification and quantification**. Raw files from DDA measurements were searched against the Uniprot mouse database using Proteome Discoverer v2.2 software (Thermo Fisher Scientific, USA). Digestion enzyme specificity was set to Trypsin/P. Precursor and fragment mass tolerance were set to 10 ppm and 0.02 Da, respectively. Modification included carbamidomethylation of cysteine as a fixed modification, with oxidation of methionine and acetyl (protein N-terminus) as variable modifications. Up to two missed cleavages were allowed. A decoy database was included to calculate the FDR. Search results were filtered with FDR 0.01 at both peptide and protein levels. Filtered output was used to generate a sample-specific spectral library using Spectronaut software (Biognosys, Switzerland). Raw files from DIA measurements were used for quantitative data extraction with the generated spectral library. FDR was estimated with the mProphet approach[71] and set to 0.01 at both peptide precursor level and protein level[72]. Data were filtered with FDR < 0.01 in at least half of the samples.

**RNA immunoprecipitation**. RIP was done on MIN6 cells using anti-CNOT3 antibody (generated by immunizing mice in cooperation with Bio Matrix Research Incorporation). MIN6 cells, 80% confluent in 10-cm dishes were lysed with 1 mL TNE buffer per dish. MIN6 cell lysates were then incubated with 1 µg of either anti-CNOT3 or control IgG antibody in an upside-down tumbling manner for 1 h at 4 °C, and then incubated with 40 µL of protein G Dynabeads (Invitrogen) in an upside-down tumbling manner for 2 h at 4 °C. Later, beads were separated using a magnetic rack and washed five times with TNE buffer. RNAs in immune complexes were isolated using Isogen II, and cDNA was generated with SuperScript Reverse Transcriptase III followed by qPCR. Abundance of each mRNA is quantified as % of input mRNA. Then mRNA bound to anti-CNOT3 antibody was presented as fold change relative to mRNA bound to control IgG antibody.

**Statistics and reproducibility**. Differences between groups were examined for statistical significance using the Student's t test (two-tailed distribution with two-sample equal variance), Mann–Whitney test, or One-way ANOVA. Analyses were performed using GraphPad Prism (GraphPad Software v.8.0), RStudio (v.1.1.453) and GSEA (v.4.0.1) developed by Broad Institute (http://software.broadinstitute.org/gsea/index.jsp). A P value of < 0.05 was considered statistically significant. Sample sizes and number of replicates are indicated in figure legends.

**Reporting summary**. Further information on research design is available in the Nature Research Reporting Summary linked to this article.

## Data availability

All source data underlying the graphs and charts presented in the main and supplementary figures are presented in Supplementary Data 1. Raw transcriptomic data sets described in the current study are available through ArrayExpress under accession number E-MTAB-8729. Mass spectrometric proteomics data have been deposited in the ProteomeXchange Consortium via the PRIDE partner repository with the dataset identifier PXD018403.

## Code availability

The scripts used for differential expression analysis of RNA-seq data and mRNA stability analysis are provided in the Supplementary Information file. The script used for mRNA stability analysis was previously published in ref. [40] and can be accessed via this link: https://static-content.springer.com/esm/art%3A10.1038%2Fnbt.3269/MediaObjects/41587_2015_BFnbt3269_MOESM34_ESM.tar. We also deposited all the scripts we used and their data files in GitHub: https://github.com/Dina-Mostafa/Scripts-for-differential-gene-expression-analysis.

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

## Acknowledgements

This work was supported by funding of the Cell Signal Unit at the Okinawa Institute of Science and Technology (OIST) Graduate University. We thank the next-generation sequencing section at OIST Graduate University for performing library preparations and RNA-seq. A.Y. was supported by Grant-in-Aid for Scientific Research (C) (18K06975) from the Japan Ministry of Education, Culture, Sports, Science and Technology. G.A.R. was supported by a Wellcome Investigator (212625/Z/18/Z) Award, MRC Programme grants (MR/R022259/1, MR/J0003042/1, MR/L020149/1), MRC (MR/N00275X/1), and Diabetes UK (BDA/11/0004210, BDA/15/0005275, BDA 16/0005485) project grants. This work has received support from the EU/EFPIA/Innovative Medicines Initiative 2 Joint Undertaking (RHAPSODY grant no 115881) to G.A.R. We thank Patrick Stoney, Akinori Takahashi and Shohei Takaoka (Cell Signal Unit, OIST) for the helpful comments and advice. We are grateful to Toshio Sasaki (Imaging section, OIST) for helping with islet sample preparation for TEM and imaging. We also thank Julie Chouinard (Neurobiology Research Unit, OIST) for helping D.M. use Imaris software.

## Author contributions

D.M. and T.Y. designed and directed the study. D.M. conducted all the experiments. A.Y. performed the bulk poly(A) tail analysis experiment, some immunohistochemistry and provided GSIS and immunohistochemistry protocols. E.G. and G.A.R. performed islet Ca$^{2+}$ imaging. Y.W. conducted the proteomic analysis. E.G. and T. Stylianides generated and used the Matlab scripts for the connectivity maps and heatmaps. A.Y. and T. Suzuki provided critical comments and suggestions and edited the figures. D.M., A.Y., G.A.R., T. Suzuki, and T.Y. wrote the manuscript with input from all authors. All authors reviewed and approved the manuscript.

## Competing interests

The authors declare no competing interests.
