## [Peer Review File · Communications Biology]

Reviewers' comments:

Reviewer #1 (Remarks to the Author):

The manuscript entitled "CNOT3-dependent mRNA deadenylation maintains β cell identity to prevent diabetes" by Mostafa and colleagues analyzes the physiological impact of the CNOT3 subunit of the CCR4-NOT deadenylation complex on the homeostasis of the pancreatic β cells.

Most of the experimental approaches involving in vivo studies are outside of this reviewer's technical competence to evaluate but my impression is that the data is carefully analyzed and described. I would, therefore, limit my comments to where I feel can make a judgment, as well as the text and the wider biological/physiological relevance.

The title can be construed as misleading in that the presence of the CNOT3 may serve a direct physiological role in preventing diabetes. However, I was left under the impression that this is somewhat overstated as authors present little direct evidence to support such a bold claim. I also feel that perhaps complementation (expression mediated by stable or transient transfection of the Cnot3 on a plasmid in KO background) is lacking in this study. Although I would not make it a condition for acceptance, it would strengthen the study if the authors could complement Cnot3 in Cnot3KO MIN6, which exhibits the diabetic phenotype. This reviewer fully recognizes that this may be technically very challenging to accomplish.

Minor comments:

Line 59: "poly(A) tail-binding proteins". PABPC1 is an acronym for the cytoplasmic polyadenosine-binding protein 1 (not a poly(A) tail-binding protein).

Line 61: "CCR4-NOT is a multimeric complex of > 2MDa". This statement may be true for the yeast Ccr4-Not but it is not precise in describing the human CCR4-NOT complex. The molecular weights of CNOT1, 2, 3, 6, 7, 9, 10 and 11 subunits (listed in the following sentence) illustrate this point as they do not add up to 2MDa. Apart from this, I would point that there are more recent reviews than the one cited here as a general reference for the CCR4-NOT complex.

Lines 223-224: "...We confirmed the upregulation of MCT1, LDHA, and HK1 at the protein level as well (Fig. 4g, h, Table 1)". Although I may have missed something, I am concerned about this panel since the images are strikingly different from other western blots, especially GAPDH control. Panels 1A, 1B and 4B appear as one would expect from a western but the bands in the 4G appear completely artificial. An explanation is perhaps necessary to clarify.

Lines 386-387: "...our work identifies CNOT3 as a key post-transcriptional regulator of β cell identity and function...". Since the study did not evaluate in detail other subunits of the CCR4-NOT complex and concentrated on CNOT3 alone, I would suggest replacing "...as a key..." with "as an important" or rephrasing more carefully in some other way. This reviewer is rather of an opinion that it is highly probable that impairment of other subunits (such as the catalytic subunits CNOT7/8 and 6/6L) will lead to a similar if not the same phenotype. The stability of the complex and indeed its function should be considered as more than the sum of interdependent parts.

Eugene Valkov, National Institutes of Health, USA.

Reviewer #2 (Remarks to the Author):

Mostafa and colleagues in their present study suggest a critical role of CNOT3 (a CCR4-NOT deadenylase complex subunit) in physiological beta-cell function and beta cell-specific

transcriptional identity. Initially, Mostafa and colleagues demonstrate CNOT1/2/3 to be dysregulated in db/db mice, with an upregulation of CNOT8. Furthermore, using a beta-cell specific CNOT3 knock out mouse model (CNOT3BetaKO) they demonstrate progressive age-dependent loss of glucose tolerance via classical IPGTT, in addition to progressive murine diabetogenesis and weight loss.

They follow-up these studies with in vitro static-GSIS and secretagogue mediated intracellular Ca⁺⁺ dynamics on CNOT3BetaKO and control derived islets. In their experiments, Mostafa and colleagues show decreased glucose dependent insulin secretion via decreased intracellular Ca⁺⁺ localization. Following up on these experiments, the authors show a decrease in Ins1/2 gene expression and suggest a decrease/loss of Ins gene expression in multiple beta-cells within the CNOT3betaKO islets, they support this finding by showing decreased insulin content, beta-cell mass and islet number/pancreas -hallmarks of islet wide beta-cell failure. They also lineage trace early beta-cells to demonstrate loss of insulin protein expression in 8-week-old CNOT3BetaKO mice. Ultrastructural analysis also demonstrates reduced density and vesicular diameter, of tertiary/secondary insulin granules.

The authors, following their findings further evaluated genes defining beta-cell de-differentiation in the mouse. Intriguingly, they find an increase in progenitor markers including Ngn3 and Aldh1a3 (protein and mRNA). Other than Mafa, most of their early beta-cell genes remained non-significant when compared to controls, loss of nuclear Mafa is also evident in their IF, a property of de-differentiating beta-cells (Nishimura et al., 2009). Progenitor-like cells have a higher level of glycolytic enzyme gene expression which was evident in case of Mct1 and Ldha (See Fig1d in: Ito and Suda, 2014: Nature Reviews Molecular Cell Biology). The authors moved on and performed bulk-RNA and bulk-protein whole transcriptome and whole proteome analysis, where they found 3847 DE genes and a correlation between protein abundance and mRNA ($R^2 = 0.64$). Intriguingly, using exonic and intronic information, the authors were able to demonstrate that 254 out of 317 'stabilized' mRNAs were amongst the DE upregulated RNAs in CNOT3BetaKO mice islets. Interestingly genes Wnt5b, Slc5a10 and Aldob were upregulated. The authors follow up their hypotheses on CNOT3-mediated-mRNA-decay showing time dependent mRNA stability of Aldob, Wnt5b, Cat and Abtb2 after actinomycinD mediated transcriptional arrest in CNOT3BetaKO islets. Finally, in a compelling manner the authors demonstrate CNOT3-RNA hybridization using an RNA immunoprecipitation assay, which corresponds again to Wnt5b and Cat amongst others.

All in all, this is a very interesting paper, showing intriguing and compelling data. It has been well written and structured. This data is of interest not only to the diabetes field but also the ever-growing field of RNA biology. Since such little is known on CNOT3, this paper enhances our critical knowledge of how CNOT proteins may orchestrate dynamic RNA regulation via RNA-decay. The authors have not only shown RNA dynamism, but corelated it to functional glucose homeostasis mediated by murine pancreatic beta-cells. As so little is known on CNOT proteins in the pancreas, it is my opinion that this manuscript has the potential to be highly cited by the field and is of interest to the broad readership of Nature Communications Biology.

Albeit, there are some major and minor deficiencies that require addressing before this paper may be considered for publication. These comments are suggested to increase the rigor of your hypothesis testing and subsequent interpretations. I have divided the deficiencies into 3 categories so that it may be easy for the authors to address them: 1) MAJOR (extremely necessary suggested changes and need addressing with experiments or additional analysis), 2) MINOR (suggested changes to the manuscript OR a simple commentary/explanation/rewording will correct this) and 3) COMMENTS (these are questions that I have to increase my own understanding of your data and the authors may choose not to respond to them, choosing not to answer certain 'comments' will have no negative impact on my decision).

MAJOR:

1. Images of Gels for this manuscript have not been included in the supplementary files, if they

are please outline where, this is necessary. Please include annotated whole-gel images in a supplementary excel file. Please perform quantitation and show this quantitation in the form of scatter plots/bar charts to ascertain relative protein expression across 3x experiments. For example, the authors claim that CNOT1/2/3 goes down in islets across 3n, this should be quantitatively shown using plots. Quantitation is necessary, because the background of the gels has been normalized differently (some gels have a whiter background while others are gray), which isn't a problem if you have shown the gel images in the supplementary file along with quantitation. This should be done for all gels in the manuscript. Such details add rigor and strengthen reproducibility of your data.

2. X-axis missing in Fig 2d, i, k and m. A key should be added in some shape or form, so that the reader doesn't have to interpret that Fig. 2g has the only key for the entire figure.

3. Please compare AUC for KCl. Add that plot/comparison as figure 2n.

4. Please add staining for an Insulin/Glucagon/CNOT3/DAPI staining to replace figure 2b. Right now, the reader has to 'guess' that only beta-cells are present in the core of islets with alpha cells in the periphery. This can be confusing as in case of humans a mosaic pattern is observed, and everyone may not be aware of species-specific differences in case of islet organization.

5. The authors can't claim that beta-cell specific insulin levels are lower based on imaging shown in Fig3b, as these cells are not lineage traced (if they are that information is lacking in the legend), it is an assumption that they are non-functional beta-cells. Please replace this with an eGFP/Insulin/DAPI merge/composite image similar to Fig2f. Please also provide quantitation, for DAPI+/eGFP+/INS+ cells in controls vs. CNOT3BetaKO derived islets.

6. Based on SEM imaging, please quantify mitochondria/beta cell and plot. It would be interesting to see if there any differences in mitochondrial percentage, based of images in Fig 3h it seems there are fewer/smaller mitochondria in CNOT3BetaKO beta cells.

7. Please quantify DAPI+/INS+/MAFA(NUCLEAR)+ and DAPI+/INS+/GLUT2+ cells in controls vs CNOT3BetaKO derived islets and plot.

8. There is no detailed coding script for the RNAseq or PROTmassspec data. Please create an account on Github and create a coding repository 'repo' where you can deposit coding used for your analysis and plot generation. This is now a norm in case of high-throughput analysis and in my opinion should be mandatory, as sequencing computational analysis is akin to running a wet-lab experiment. I was not able to evaluate any of your coding and analysis workflows because I didn't have access to your R/python/linux scripts. You may create a private repo in Github, till your manuscript is accepted, but it is necessary. You can look at how to make a repo here: <https://help.github.com/en/github/getting-started-with-github/create-a-repo>; for an example please look here: https://github.com/Dragonmasterx87/Pancreas_ductal_scRNAseq). You don't need something very fancy, just a description and your code will suffice. I will parse through your code once you re-submit revisions.

9. Since the coding script is not available, I was not able to comment on the statistical models used for RNAseq analysis, please include in the revisions so that I may comment.

10. Have the authors completed a sequence deposition in NCBI GEO or EMBL-EBI? Have the authors completed a PRIDE ARCHIVE submission for protein mass-spec information? This is mandatory in my opinion and is generally a good practice. The quantitative data in this manuscript cannot be utilized by your fellow colleagues unless it carefully curated by NCBI or EMBL-EBI. The excellent data your manuscript provides should be made available to your colleagues to further your findings. As the computational biologists in this manuscript will be aware, all their data analysis can be replicated purely from FASTQ files which can be easily converted from sequence

repository archive files deposited in NCBI GEO (an example). Most repositories allow your data to remain embargoed until your paper is published/accepted.

11. Please add a PCA plot of the sequenced samples 3 + 3 so that a whole transcriptome comparison can be made, your experimental samples should cluster together as should your controls. You may include this information in your supplementary file. I am asking for a PCA plot because there is no heatmap of the top 500 most variable genes, so I am guessing the bulk-RNAseq shows sample-conserved differences.

12. The interpretation, that a CNOT3 is a direct target of 80% of stabilized RNA, is a very indirect measure of loss-of-CNOT3-mediated-RNA-decay on RNA stability. Such massive RNA regulation could also be possible due to downstream effects of CNOT3 mediated RNA decay on other key regulatory proteins and miRNA. Theoretically, if stabilized RNA in the CNOT3BetaKO model correlated to upregulated RNA so tightly then how do the authors explain the remaining ~20% or 63 genes? I appreciate, that this is an interesting mathematical application of Gaidatzis et al., 2015's model to evaluate the efficiency of RNA processing, unfortunately it alone is not enough to suggest direct correlation. The authors should at the very least appreciate limitations of this analysis in their discussion.

13. As this deletion of CNOT3 is also occurring in INS+ cells within the arcuate nucleus, ventromedial nucleus and median area eminence in the hypothalamus, what is the food intake of these mice? Have the authors looked at CNOT expression in the hypothalamus? This is a limitation of the Cre-tracers used currently, and if the authors have not performed the analysis, they must comment on this. Please see Fig 1 Schwartz et al., 2010: Diabetes.

MINOR:

1. In addition to point 12 (MAJOR comments), can the authors make a correlation comparing maybe the top 50 (or select based on some other parameter) and compare RNA stabilization-Upregulated RNAs-protein abundance comparing CNOT3BetaKO and controls? This would be interesting to show, that these fluctuations in RNA transcriptional dynamics have a direct correlation to protein translation purely due to RNA abundance. Since the authors have the data...it would be interesting to investigate. Such a comparative analysis can be added to the supplemental information.

2. I find the RIP assay to be very intriguing, it would be interesting to perform a RNAseq experiment on CNOT3-RIP isolated RNA (RIPseq please see Zheng et al., 2018: Plos Biology). I have added this suggestion in the minor section, as I don't feel at this moment it is necessary for the manuscript, but I suggest if the authors can, they should attempt this. In my opinion this will comprehensively show CNOT3 RNA targeting. This would be a very compelling experiment, as you could then show preferential RNA targeting and correlate that to intronic and exonic RNA species, if not the authors should comment on this approach in a limitations section. RNA decay dynamics vary for different RNA species and such findings would be interesting (please see Garneau et al., 2007: Nature Reviews Molecular Cell Biology; Raisch et al., 2019: Nature Communications; Stowell et al., 2016: Cell Reports)

3. Did the authors use 2g/gBW glucose in their IPGTT? Is this a typo? Isn't classical IPGTT 2g/KgBW? (Please see: <https://www.mmpc.org/shared/document.aspx?id=238&docType=Protocol>).

4. Literature suggests, CNOT3 may also be associated with progenitor maintenance and proliferation, however in their manuscript the authors suggest that CNOT3 actually promotes the maintenance of a functional beta-cell (Zheng et al., 2016: Stem Cell Reports; Zheng et al., 2013: Stem Cells; Zhou et al., 2017: Scientific Reports) and that loss of CNOT actually leads to a de-

differentiated beta cell. Alternatively, it appears that CNOT3 along with BMP and FGF signaling initiates mesendodermal differentiation in ES cells (Sarkar et al., 2019: bioRxiv) this could be interesting, as the mesendoderm forms the endoderm, marking an important step in the highway to pancreatic endocrinogenesis. The authors should please, succinctly discuss previous CNOT3 functions, in the context of their findings.

5. Did the authors add BSA to their Krebs ringer buffer and normalize it to a pH of 7.4? usually the buffer is a little acidic (~pH6.8) and needs to be equilibrated to a pH of 7.4. This is necessary to mention as in my experience mouse islets have different secretion dynamics in a pH of 7.4 vs. 6.8-6.9. They should mention this in their methods, they don't need to re-do their experiments in 7.4 if they didn't, but they should report it in the methods so that someone attempting to re-produce their data will be able to robustly.

6. The authors should please normalize their secretion data to total insulin content and add that data to the supplemental figures. I feel an insulin-based normalization would be interesting in the context of fewer functional beta cells, in CNOT3BetaKO mice. I am interested in knowing what the few 'functional' beta-cells in these CNOT3BetaKO mice are in comparison to controls, this is not possible right now based purely on genomic DNA normalization.

7. Many 'n' are missing for certain experiments and should be included (for example Fig 2c, a description of pooling is shown but not how many times the experiment was repeated).

COMMENTS:

1. What is the upper limit for the Glutest Pro glucometer? I was surprised that some of your values for the GTT were beyond 600mg/dl (Fig 2g), is this device accurate beyond 600mg/dl?

2. It seems that most of the pathways you have picked in the GO analysis (upregulated) are canonical pathways of cell signaling and cellular immune response. Did you not see any pathways that support your hypothesis of de-differentiation? Maybe it would be better to include that in your graph.

3. Since this is an interesting paper, I suggest making a simple diagram demonstrating how CNOT3 functions in a beta-cell and how its lacking adversely affects a beta-cell. It would be nice to include that as a final figure (or in Fig6?), or even as a graphical abstract.

4. I have placed this as a comment, as it is a personal observation but the thickness of the lines in many of the graphs are different from one another, will it be possible to make all the line thickness uniform in all of the figures? I would appreciate the uniformity and it will make your graphics more appealing.

5. Why did you choose RIP over RNA-CLIP?

FINAL COMMENTS:

As Nature Communications Biology allows authors to sign reviews I am doing so. I believe if the authors were respectful enough to ask another colleague's opinion on their work (without hiding their identity), then the same should be reciprocated. I appreciate Nature Communications Biology on making the review process transparent and open. Please note, that these comments reflect my personal opinions as a scientist, and do not reflect the opinions and views of my previous or present labs. I look forward to reading a revised version of their manuscript.

Best of luck,
Mirza Muhammad Fahd Qadir
Post-Doctoral Fellow
Tulane University Health Sciences Center
1430 Tulane Ave.

Reviewer #3 (Remarks to the Author):

This study by Mostafa et al describes studies on dysregulation of the CCR4-NOT deadenylase complex in diabetes. The study demonstrates that beta cell targeted deletion of the Cnot3 subunit of the CCR4-NOT complex in mice led to impaired glucose tolerance, decreased β cell mass, and development of diabetes. Cnot3 β KO islets displayed altered deadenylation and increased mRNA stability of beta cell-disallowed genes and genes relevant to altered beta cell function. These results suggested the novel finding that CNOT3-mediated mRNA deadenylation and decay constitute post-transcriptional mechanisms essential for β cell identity. This important study is well done, and should be of interest to a wide readership.

My major criticism is the failure to fully discuss results of the RNA-seq results in Figure 5, where gene expression changes between Cnot3 β KO and control islets were measured. As shown in Figure 5, the major functional processes enriched in genes up-regulated by Cnot3 disruption were immune and interferon stimulated genes. This is unexpected, at least to me, because the mouse model used was one of type 2, or supposed non-immune diabetes. I think this result is important because the widely cited boundaries between types 1 and 2 diabetes are becoming increasingly blurred. For instance, recent studies show that T1D may often be mis-diagnosed as T2D, especially in older subjects (*Diabetologia* volume 62, pages1167–1172(2019)). On a higher level, the authors' results showing immune and interferon-related gene changes on disruption of the CCR4-NOT complex suggest that this complex may be a molecular link between the pathologies of T1D and T2D. I think this concept is certainly worthy of discussion.

First of all, we would like to thank the reviewers for the constructive feedback. We responded to the comments point by point and added the necessary changes in the manuscript. We would like to point out that we replaced Fig. 6c with the revised one as we recalculated the relative mRNA abundance (%) in Act. D-treated samples (4h or 8h) compared to that in untreated samples (0h). The values in untreated samples (0h) are set as 100.

Reviewer #1 (Remarks to the Author):

The manuscript entitled “CNOT3-dependent mRNA deadenylation maintains β cell identity to prevent diabetes” by Mostafa and colleagues analyzes the physiological impact of the CNOT3 subunit of the CCR4-NOT deadenylation complex on the homeostasis of the pancreatic β cells.

Most of the experimental approaches involving *in vivo* studies are outside of this reviewer’s technical competence to evaluate but my impression is that the data is carefully analyzed and described. I would, therefore, limit my comments to where I feel can make a judgment, as well as the text and the wider biological/physiological relevance.

The title can be construed as misleading in that the presence of the CNOT3 may serve a direct physiological role in preventing diabetes. However, I was left under the impression that this is somewhat overstated as authors present little direct evidence to support such a bold claim. I also feel that perhaps complementation (expression mediated by stable or transient transfection of the *Cnot3* on a plasmid in KO background) is lacking in this study. Although I would not make it a condition for acceptance, it would strengthen the study if the authors could complement *Cnot3* in *Cnot3*KO MIN6, which exhibits the diabetic phenotype. This reviewer fully recognizes that this may be technically very challenging to accomplish.

We would like to thank reviewer #1 for the constructive feedback. We have changed the manuscript title to “Disruption of CNOT3-dependent mRNA deadenylation leads to loss of β -cell identity and a diabetic phenotype.” We agree that reintroduction of CNOT3 into *Cnot3*-KO backgrounds (mice and MIN6 cells) would strengthen the importance of CNOT3 in maintaining β cell identity. Since we aim to understand roles of mRNA deadenylation in β cell development and function *in vivo*, in future work we plan to perform rescue experiments reintroducing CNOT3 effectively into β -cells from *Cnot3* β KO mice.

Minor comments:

Line 59: “poly(A) tail-binding proteins”. PABPC1 is an acronym for the cytoplasmic polyadenosine-binding protein 1 (not a poly(A) tail-binding protein).

We replaced poly(A) tail-binding proteins with cytoplasmic polyadenosine-binding protein 1 (PABPC1) (line 59 in the revised manuscript).

Line 61: “CCR4–NOT is a multimeric complex of > 2MDa”. This statement may be true for the yeast Ccr4-Not but it is not precise in describing the human CCR4-NOT complex. The molecular

weights of CNOT1, 2, 3, 6, 7, 9, 10 and 11 subunits (listed in the following sentence) illustrate this point as they do not add up to 2MDa. Apart from this, I would point that there are more recent reviews than the one cited here as a general reference for the CCR4-NOT complex.

We deleted the complex molecular weight information, because it was not necessary in the present work. In addition, we replaced the review by Collart and Panasenko (2012) with a more recent review (Collart et al, 2016) as reference #18 in line 874 in the revised manuscript.

Lines 223-224: "... We confirmed the upregulation of MCT1, LDHA, and HK1 at the protein level as well (Fig. 4g, h, Table 1)". Although I may have missed something, I am concerned about this panel since the images are strikingly different from other western blots, especially GAPDH control. Panels 1A, 1B and 4B appear as one would expect from a western but the bands in the 4G appear completely artificial. An explanation is perhaps necessary to clarify.

The immunoblot images in this manuscript, except for Fig. 4g, were generated using conventional SDS-PAGE followed by immunoblot analyses. Fig. 4g was generated using Simple-Wes, which provides images that are different from the conventional method. The reason that we used Simple-Wes in Fig. 4g was already described in Methods (line 461-462). Raw data files are added as supplemental information.

Lines 386-387: "...our work identifies CNOT3 as a key post-transcriptional regulator of β cell identity and function...". Since the study did not evaluate in detail other subunits of the CCR4-NOT complex and concentrated on CNOT3 alone, I would suggest replacing "...as a key..." with "as an important" or rephrasing more carefully in some other way. This reviewer is rather of an opinion that it is highly probable that impairment of other subunits (such as the catalytic subunits CNOT7/8 and 6/6L) will lead to a similar if not the same phenotype. The stability of the complex and indeed its function should be considered as more than the sum of interdependent parts.

We agree with this comment. We assume that the CCR4-NOT complex as a functional entity, rather than CNOT3 alone, plays critical roles in β cells. While CNOT3 is important for stability and function of the complex, other subunits also contribute to complex formation and mRNA deadenylation. As suggested, we replaced "as a key" with "as an important" in line 407.

Eugene Valkov, National Institutes of Health, USA.

Reviewer #2 (Remarks to the Author):

MAJOR:

1. Images of Gels for this manuscript have not been included in the supplementary files, if they are please outline where, this is necessary. Please include annotated whole-gel images in a supplementary excel file. Please perform quantitation and show this quantitation in the form of scatter plots/bar charts to ascertain relative protein expression across 3x experiments. For example, the authors claim that CNOT1/2/3 goes down in islets across 3n, this should be quantitatively shown using plots. Quantitation is necessary, because the background of the gels has been normalized differently (some gels have a whiter background while others are gray), which isn't a problem if you have shown the gel images in the supplementary file along with quantitation. This should be done for all gels in the manuscript. Such details add rigor and strengthen reproducibility of your data.

We provided uncropped, annotated gel images of all immunoblot analyses in the supplementary files. We quantified all Western blot and Simple Wes images and showed the results as scatter plots. Please note that we cut the membranes and probed each piece of the membrane separately with different antibodies, thus membrane sizes vary among the experiments. Based on quantification results, we amended some parts of the main text in the Results section (lines 89-103) and Discussion section (lines 379-384). We replaced Fig.1c with more representative figure according to quantitative data.

2. X-axis missing in Fig 2d, i, k and m. A key should be added in some shape or form, so that the reader doesn't have to interpret that Fig. 2g has the only key for the entire figure.

We appreciate this point. We added the missing X-axis label to Fig. 2d. We also added keys to Figures 2h-2k.

3. Please compare AUC for KCl. Add that plot/comparison as figure 2n.

Following the reviewer's suggestion, we compared the AUC under KCl, and included this as Fig. 2n. We also modified the corresponding text in line 145 in the revised manuscript. The method of AUC calculation is described in lines 506-509 and Fig. 2 legend (lines 751-753).

4. Please add staining for an Insulin/Glucagon/CNOT3/DAPI staining to replace figure 2b. Right now, the reader has to 'guess' that only beta-cells are present in the core of islets with alpha cells in the periphery. This can be confusing as incase of humans a mosaic pattern is observed, and everyone may not be aware of species-specific differences in case of islet organization.

We changed the description of Fig. 2b in lines 108-113 in the revised manuscript so that readers can understand which cells lack CNOT3 expression and which do not in *Cnot3* β KO panel. As can be seen in Fig.3g, Cre-mediated recombination occurs exclusively in β cells but not in

peripherally located non- β islet cells expressing GLUC, SST and PPT. We think that these results are sufficient to indicate β -cell specific CNOT3 suppression in islets.

5. The authors can't claim that beta-cell specific insulin levels are lower based on imaging shown in Fig3b, as these cells are not lineage traced (if they are that information is lacking in the legend), it is an assumption that they are non-functional beta-cells. Please replace this with an eGFP/Insulin/DAPI merge/composite image similar to Fig2f. Please also provide quantitation, for DAPI+/eGFP+/INS+ cells in controls vs. CNOT3BetaKO derived islets.

As suggested, we replaced the original image with one showing EGFP/insulin/DAPI staining. We counted the number of INS+ cells among EGFP+ cells and compared the value between control and *Cnot3* β KO islets (Fig.3b, c in the revised manuscript). We obtained the data from three biological replicates for each genotype (n=3).

6. Based on SEM imaging, please quantify mitochondria/beta cell and plot. It would be interesting to see if there any differences in mitochondrial percentage, based of images in Fig 3h it seems there are fewer/smaller mitochondria in CNOT3BetaKO beta cells.

The size and number of mitochondria appeared to vary, even in β cells in control islets. Indeed, there was no significant difference in mitochondrial number between genotypes. We thought that only a single set of images would be misleading and added more images of control and *Cnot3* β KO islets to Supplementary Fig. 4.

7. Please quantify DAPI+/INS+/MAFA(NUCLEAR)+ and DAPI+/INS+/GLUT2+ cells in controls vs CNOT3BetaKO derived islets and plot.

We quantified DAPI+/INS+/MAFA(NUCLEAR)+ and DAPI+/INS+/GLUT2+ cells in control and *Cnot3* β KO islets and included corresponding plots in Supplementary Fig. 5.

8. There is no detailed coding script for the RNAseq or PROTmassspec data. Please create an account on Github and create a coding repository 'repo' where you can deposit coding used for your analysis and plot generation. This is now a norm in case of high-throughput analysis and in my opinion should be mandatory, as sequencing computational analysis is akin to running a wet-lab experiment. I was not able to evaluate any of your coding and analysis workflows because I didn't have access to your R/python/linux scripts. You may create a private repo in Github, till your manuscript is accepted, but it is necessary. You can look at how to make a repo here: <https://help.github.com/en/github/getting-started-with-github/create-a-repo>; for an example please look here: https://github.com/Dragonmasterx87/Pancreas_ductal_scRNAseq). You don't need something very fancy, just a description and your code will suffice. I will parse through your code once you re-submit revisions.

As described in the Methods section (lines 601-610), paired-end RNA-seq data were mapped to the *Mus musculus* reference strain mm10 UCSC using StrandNGS, next generation sequencing analysis software (Strand Genomics, Inc.). The resultant raw count data were analyzed. We used the edgeR function of Robinson et al. (2010) for differential expression analysis. The script we used is adapted from the script in the book *Basic Applied Bioinformatics* by Chandra Sekhar Mukhopadhyay, Ratan Kumar Choudhary and Mir Asif Iquebal. For evaluation purposes, we are providing the codes as text files. As for the analysis of proteomic data, we didn't use a coding script, but Proteome Discoverer v2.2 software, as described in the Methods section in the modified manuscript, line 661.

9. Since the coding script is not available, I was not able to comment on the statistical models used for RNAseq analysis, please include in the revisions so that I may comment.

Please refer to our response to your major point 8.

10. Have the authors completed a sequence deposition in NCBI GEO or EMBL-EBI? Have the authors completed a PRIDE ARCHIVE submission for protein mass-spec information? This is mandatory in my opinion and is generally a good practice. The quantitative data in this manuscript cannot be utilized by your fellow colleagues unless it carefully curated by NCBI or EMBL-EBI. The excellent data your manuscript provides should be made available to your colleagues to further your findings. As the computational biologists in this manuscript will be aware, all their data analysis can be replicated purely from FASTQ files which can be easily converted from sequence repository archive files deposited in NCBI GEO (an example). Most repositories allow your data to remain embargoed until your paper is published/accepted.

We deposited our RNA-seq data in Arrayexpress under accession number: **E-MTAB-8729**, and proteomics data to the ProteomeXchange Consortium via the PRIDE partner repository with data set identifier **PXD018403**. We included a statement about data availability in the revised manuscript, in lines 691-697.

The data is private, and we plan to make it public after the acceptance of our manuscript. Kindly find bellow access information for reviewer:

Access information for data on Arrayexpress:

Username: Reviewer_E-MTAB-8729

Password: abVbiph4

Access information for data on PRIDE:

Username: reviewer01967@ebi.ac.uk

Password: n2roOFCv

11. Please add a PCA plot of the sequenced samples 3 + 3 so that a whole transcriptome comparison can be made, your experimental samples should cluster together as should your

controls. You may include this information in your supplementary file. I am asking for a PCA plot because there is no heatmap of the top 500 most variable genes, so I am guessing the bulk-RNAseq shows sample-conserved differences.

We added a multidimensional scaling (MDS) plot, which is equivalent to PCA in Supplementary Fig. 6. *Cnot3*βKO samples cluster separately from control samples.

12. The interpretation, that a CNOT3 is a direct target of 80% of stabilized RNA, is a very indirect measure of loss-of-CNOT3-mediated-RNA-decay on RNA stability. Such massive RNA regulation could also be possible due to downstream effects of CNOT3 mediated RNA decay on other key regulatory proteins and miRNA. Theoretically, if stabilized RNA in the CNOT3BetaKO model correlated to upregulated RNA so tightly then how do the authors explain the remaining ~20% or 63 genes? I appreciate, that this is an interesting mathematical application of Gaidatzis et al., 2015's model to evaluate the efficiency of RNA processing, unfortunately it alone is not enough to suggest direct correlation. The authors should at the very least appreciate limitations of this analysis in their discussion.

We agree that these 80% stabilized mRNAs are not necessarily CNOT3 direct targets. Using this mathematical method, we cannot know whether genes are directly or indirectly affected by CNOT3 suppression. We amended the sentence to simply explain our results as follows: We found that 80% (254 of 317 mRNAs) of mRNAs that show increased Δexon-Δintron in *Cnot3*βKO compared to control islets were among the significantly upregulated mRNAs, suggesting that post-transcriptional mechanisms, including mRNA stabilization, effectively contributed to an increase of mRNAs in *Cnot3*βKO islets. In contrast, decreased intron reads in *Cnot3*βKO islets were not relevant to any significant increase of 63 mRNAs, despite increased Δexon-Δintron (lines 261-266).

13. As this deletion of CNOT3 is also occurring in INS+ cells within the arcuate nucleus, ventromedial nucleus and median area eminence in the hypothalamus, what is the food intake of these mice? Have the authors looked at CNOT expression in the hypothalamus? This is a limitation of the Cre-tracers used currently, and if the authors have not performed the analysis, they must comment on this. Please see Fig 1 Schwartz et al., 2010: Diabetes.

We understand this limitation caused by *Ins2-cre* tracers. Therefore, we used an *Ins1-cre* model to overcome this limitation. *Ins1-cre* mice do not exhibit Cre-mediated recombination in the brain¹⁻³.

- 1 Hasegawa, Y. *et al.* Generation and characterization of *Ins1-cre*-driver C57BL/6N for exclusive pancreatic beta cell-specific Cre-loxP recombination. *Exp Anim* **63**, 183-191, doi:10.1538/expanim.63.183 (2014).
- 2 Hasegawa, Y. *et al.* Generation of CRISPR/Cas9-mediated bicistronic knock-in *Ins1-cre* driver mice. *Experimental Animals* **65**, 319-327, doi:10.1538/expanim.16-0016 (2016).

- 3 Wicksteed, B. *et al.* Conditional gene targeting in mouse pancreatic β -cells: analysis of ectopic Cre transgene expression in the brain. *Diabetes* **59**, 3090-3098, doi:10.2337/db10-0624 (2010).

MINOR:

1. In addition to point 12 (MAJOR comments), can the authors make a correlation comparing maybe the top 50 (or select based on some other parameter) and compare RNA stabilization-Upreregulated RNAs-protein abundance comparing CNOT3BetaKO and controls? This would be interesting to show, that these fluctuations in RNA transcriptional dynamics have a direct correlation to protein translation purely due to RNA abundance. Since the authors have the data...it would be interesting to investigate. Such a comparative analysis can be added to the supplemental information.

The top 50 mRNAs showing both upregulation and stabilization are not necessarily detected in MS analysis. We found that only 71 genes from a total of 254 significantly upregulated, stabilized genes are detected as significantly upregulated proteins (≥ 1.5 -fold). But there was no significant correlation between their differential gene expression (TR_log₂FC) and differential protein expression (MS_log₂FC). Consequently, we didn't include this analysis results.

2. I find the RIP assay to be very intriguing, it would be interesting to perform a RNAseq experiment on CNOT3-RIP isolated RNA (RIPseq please see Zheng et al., 2018: Plos Biology). I have added this suggestion in the minor section, as I don't feel at this moment it is necessary for the manuscript, but I suggest if the authors can, they should attempt this. In my opinion this will comprehensively show CNOT3 RNA targeting. This would be a very compelling experiment, as you could then show preferential RNA targeting and correlate that to intronic and exonic RNA species, if not the authors should comment on this approach in a limitations section. RNA decay dynamics vary for different RNA species and such findings would be interesting (please see Garneau et al., 2007: Nature Reviews Molecular Cell Biology; Raisch et al., 2019: Nature Communications; Stowell et al., 2016: Cell Reports)

Thank you very much for this helpful comment. We will try the CNOT3-RIPseq experiment to comprehensively understand target mRNAs for the CCR4-NOT complex in β -cells in the future.

3. Did the authors use 2g/gBW glucose in their IPGTT? Is this a typo? Isn't classical IPGTT 2g/KgBW? (Please see: <https://www.mmmpc.org/shared/document.aspx?id=238&docType=Protocol>).

We used 2g/kg BW glucose in IPGTT, but we wrote it as 2 "mg/g" BW glucose in the submitted manuscript because that is an appropriate unit of mass for mice. Moreover, because both sides of the ratio are divided by 1,000, the number is unchanged. However, since 2g/kg is the most

common way of describing the glucose dose for this assay in literature, we rewrote it as 2g/kg in lines 475, 481 and 742 in the revised manuscript.

4. Literature suggests, CNOT3 may also be associated with progenitor maintenance and proliferation, however in their manuscript the authors suggest that CNOT3 actually promotes the maintenance of a functional beta-cell (Zheng et al., 2016: Stem Cell Reports; Zheng et al., 2013: Stem Cells; Zhou et al., 2017: Scientific Reports) and that loss of CNOT actually leads to a de-differentiated beta cell. Alternatively, it appears that CNOT3 along with BMP and FGF signaling initiates mesendodermal differentiation in ES cells (Sarkar et al., 2019: bioRxiv) this could be interesting, as the mesendoderm forms the endoderm, marking an important step in the highway to pancreatic endocrinogenesis. The authors should please, succinctly discuss previous CNOT3 functions, in the context of their findings.

We discussed the relation between previously reported CNOT3 functions and our present findings (lines 392-406). We explained that the CCR4–NOT complex contributes to proper mRNA expression by degrading inappropriate mRNAs, depending on the context. Thus, we are not surprised that it is associated with progenitor maintenance and proliferation, and at the same time, it is essential for tissue function/identity.

5. Did the authors add BSA to their Krebs ringer buffer and normalize it to a pH of 7.4? usually the buffer is a little acidic (~pH6.8) and needs to be equilibrated to a pH of 7.4. This is necessary to mention as in my experience mouse islets have different secretion dynamics in a pH of 7.4 vs. 6.8-6.9. They should mention this in their methods, they don't need to re-do their experiments in 7.4 if they didn't, but they should report it in the methods so that someone attempting to reproduce their data will be able to robustly.

The information was missing in the submitted manuscript. We added BSA to Kreb's Ringer buffer and the pH was 7.4. We described this in the Methods section (line 487-488) in the revised manuscript.

6. The authors should please normalize their secretion data to total insulin content and add that data to the supplemental figures. I feel an insulin-based normalization would be interesting in the context of fewer functional beta cells, in CNOT3BetaKO mice. I am interested in knowing what the few 'functional' beta-cells in these CNOT3BetaKO mice are in comparison to controls, this is not possible right now based purely on genomic DNA normalization.

As suggested, we normalized insulin secretion with total insulin content. The normalized value in islets from *Cnot3* β KO mice was higher than that from control mice, suggesting that insulin secretion is not affected in the absence of CNOT3. However, it is unclear whether the normal insulin release mechanism functions, because insulin content is very low and expression of many molecules involved in insulin release is dysregulated in *Cnot3* β KO islets as shown by RNA-seq, qPCR and proteome analysis.

7. Many 'n' are missing for certain experiments and should be included (for example Fig 2c, a description of pooling is shown but not how many times the experiment was repeated).

The bulk poly(A) tail experiment was done only once (n=1) on islets pooled from four mice. This assay requires a large amount of RNA. We got it to work only after pooling islets from 4 mice and extracting RNA. We have described this in Methods. We have already provided "n" in all other Figures.

COMMENTS:

1. What is the upper limit for the Glutest Pro glucometer? I was surprised that some of your values for the GTT were beyond 600mg/dl (Fig 2g), is this device accurate beyond 600mg/dl?

The upper limit was 600 mg/dL. At 12 weeks of age, fasting blood glucose of *Cnot3*βKO mice was very high before glucose injection. We agree that the values above 600 mg/dL might be unreliable. Accordingly, we replaced the GTT test result at 12 weeks old with fasting blood glucose levels.

2. It seems that most of the pathways you have picked in the GO analysis (upregulated) are canonical pathways of cell signaling and cellular immune response. Did you not see any pathways that support your hypothesis of de-differentiation? Maybe it would be better to include that in your graph.

We couldn't find any GO terms relevant to dedifferentiation or immature state among significantly enriched terms using the DAVID gene annotation tool. It should be noted that we don't claim dedifferentiation per se, but incomplete maturation and failure to achieve a fully differentiated state, as described in line 191. We used the progenitor/dedifferentiation markers only as indicators for the developmental defect and incomplete maturation of β cells.

3. Since this is an interesting paper, I suggest making a simple diagram demonstrating how CNOT3 functions in a beta-cell and how its lacking adversely affects a beta-cell. It would be nice to include that as a final figure (or in Fig6?), or even as a graphical abstract.

We added a schematic model of CCR4-NOT function in β-cells in Figure 7.

4. I have placed this as a comment, as it is a personal observation but the thickness of the lines in many of the graphs are different from one another, will it be possible to make all the line thickness uniform in all of the figures? I would appreciate the uniformity and it will make your graphics more appealing.

We made the line thickness uniform across all the figures.

5. Why did you choose RIP over RNA-CLIP?

In general, RNA-CLIP works efficiently with RNA binding proteins (RBPs). CCR4-NOT complex subunits, including CNOT3, don't bind to mRNAs directly, as RBPs mediate the interaction between the CCR4-NOT complex and mRNAs, indicating that the recognition motifs depend largely on RBPs. Of course, it is possible that RNA-CLIP for molecules that indirectly bind to mRNAs including the CCR4-NOT complex will give us important information. We will plan RNA-CLIP in a future study.

FINAL COMMENTS:

As Nature Communications Biology allows authors to sign reviews I am doing so. I believe if the authors were respectful enough to ask another colleague's opinion on their work (without hiding their identity), then the same should be reciprocated. I appreciate Nature Communications Biology on making the review process transparent and open. Please note, that these comments reflect my personal opinions as a scientist, and do not reflect the opinions and views of my previous or present labs. I look forward to reading a revised version of their manuscript.

Best of luck,
Mirza Muhammad Fahd Qadir
Post-Doctoral Fellow
Tulane University Health Sciences Center
1430 Tulane Ave.
New Orleans LA 70112

Thank you very much, Dr. Qadir, for your openness! We greatly appreciate your professionalism!

Reviewer #3 (Remarks to the Author):

This study by Mostafa et al describes studies on dysregulation of the CCR4-NOT deadenylase complex in diabetes. The study demonstrates that beta cell targeted deletion of the Cnot3 subunit of the CCR4-NOT complex in mice led to impaired glucose tolerance, decreased β cell mass, and development of diabetes. Cnot3 β KO islets displayed altered deadenylation and increased mRNA stability of beta cell-disallowed genes and genes relevant to altered beta cell function. These results suggested the novel finding that CNOT3-mediated mRNA deadenylation and decay constitute post-transcriptional mechanisms essential for β cell identity. This important study is well done, and should be of interest to a wide readership.

My major criticism is the failure to fully discuss results of the RNA-seq results in Figure 5, where gene expression changes between Cnot3 β KO and control islets were measured. As shown in Figure 5, the major functional processes enriched in genes up-regulated by Cnot3 disruption were immune and interferon stimulated genes. This is unexpected, at least to me, because the mouse model used was one of type 2, or supposed non-immune diabetes. I think this result is important because the widely cited boundaries between types 1 and 2 diabetes are becoming

increasingly blurred. For instance, recent studies show that T1D may often be mis-diagnosed as T2D, especially in older subjects (Diabetologia volume 62, pages1167–1172(2019). On a higher level, the authors' results showing immune and interferon-related gene changes on disruption of the CCR4-NOT complex suggest that this complex may be a molecular link between the pathologies of T1D and T2D. I think this concept is certainly worthy of discussion.

We thank the reviewer for this good suggestion. Our findings revealed CNOT3 as an important regulator of β cell maturation. Thus, defects in β cell maturation in *Cnot3* β KO mice resulted in β cell dysfunction and in a diabetic phenotype in *Cnot3* β KO mice. We are also concerned with the contribution of altered expression of immune-related genes to the diabetic phenotypes of our *Cnot3* β KO mice. We think that our *Cnot3* β KO mouse model may recapitulate both T1D and T2D. Of course, it is possible that changes in immune-related genes affect β cell function in *Cnot3* β KO mice, especially because T1D is an autoimmune disease. But we were unable to prove the link between aberrant immune-related genes and the diabetic phenotype in our mouse model. We agree that the boundaries between T1D and T2D are becoming increasingly blurred, exemplified by “LADA”, which is phenotypically similar to T2D, but involves autoimmunity. Moreover, immune-related genes are enriched in human T2D islets and rodent models fed with a high-fat diet (<https://www.ncbi.nlm.nih.gov/pubmed/29185012>). This made us more prudent in drawing conclusions; therefore, we carefully and briefly discussed these matters in lines 384-391.

REVIEWERS' COMMENTS:

Reviewer #1 (Remarks to the Author):

I'm satisfied with the authors' response to my criticisms.

Reviewer #2 (Remarks to the Author):

This reviewer appreciates and recognizes the in-depth responses and revisions across the manuscript. Particularly in view of the current pandemic, it is notable that the authors have worked very hard to address my concerns and add additional analysis and data to support their claims.

It is interesting that insulin normalization shows that surviving beta-cells are functional. An intriguing observation, raising interesting questions for the compensatory mechanisms protecting surviving beta-cells.

I have parsed through the coding and it appears to be well-annotated for any R environment using R v3.5.2 and above. I was also able to login and view their raw data in Pride and Arrayexpress. For a few samples, the MDS plot is an appropriate 2D non-linear multidimensional analysis extrapolation. I recommend submitting your coding as a supplemental file.

I do NOT require any further revisions or responses. Submission of appropriate supplemental data (including the coding rubric) can be handled based on recommendations outlined by the managing editor of this manuscript.

Congratulations on your findings. I look forward to reading your formatted manuscript in Nature Communications Biology.

Stay safe.

Mirza Muhammad Fahd Qadir
Post-Doctoral Fellow
Tulane University Health Sciences Center
1430 Tulane Ave.
New Orleans LA 70112

Reviewer #3 (Remarks to the Author):

The authors revisions have satisfied my original concerns. I have no further objections to publication.